# ACTIVE FINE-TUNING OF GENERALIST POLICIES

## ABSTRACT

Pre-trained generalist policies are rapidly gaining relevance in robot learning due to their promise of fast adaptation to novel, in-domain tasks. This adaptation often relies on collecting new demonstrations for a specific task of interest and applying imitation learning algorithms, such as behavioral cloning. However, as soon as several tasks need to be learned, we must decide *which tasks should be demonstrated and how often?* We study this multi-task problem and explore an interactive framework in which the agent *adaptively* selects the tasks to be demonstrated. We propose AMF (Active Multi-task Fine-tuning), an algorithm to maximize multi-task policy performance under a limited demonstration budget by collecting demonstrations yielding the largest information gain on the expert policy. We derive performance guarantees for AMF under regularity assumptions and demonstrate its empirical effectiveness to efficiently fine-tune neural policies in complex and high-dimensional environments.

## 1 INTRODUCTION

The availability of large pre-trained models has transformed entire areas of machine learning, from computer vision (Krizhevsky et al., 2012; He et al., 2016; Dosovitskiy et al., 2021; Radford et al., 2021), to natural language processing (Radford et al., 2019; Brown et al., 2020) and generative modeling in general (Ho et al., 2020; Esser et al., 2024). This paradigm has started to extend to robotics and control (Collaboration, 2023; Ma et al.,

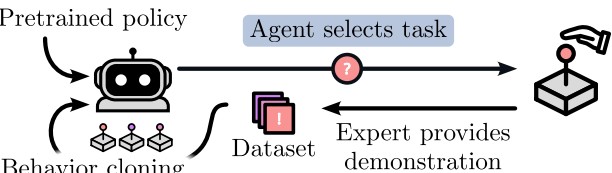

Figure 1: Interactive loop between agent and expert. We consider a scenario where we receive a pre-trained policy, and are able to obtain expert demonstrations of tasks. We study how to select tasks (in blue) to obtain the best-performing policy after as few demonstrations as possible.

2024), in particular for systems for which demonstrations are readily available (Octo Model Team et al., 2024), or can be easily collected (Zhao et al., 2023). Even when demonstrations are not easily obtained, scaling laws in reinforcement learning (Ceron et al., 2024b;a; Nauman et al., 2024) suggest the possibility of leveraging large pre-trained policies. These "generalist" policies have decent performance on many tasks, and can be fine-tuned on particular set of tasks while leveraging their previously learned representations and skills. We investigate whether representations of such policies can be used to significantly bootstrap learning progress.

As a motivating example, consider a household robot that is delivered with a pre-trained "generalist" policy, and deployed in slightly different conditions than those observed in its training data. While the robot may achieve some tasks in a zero-shot fashion (e.g. simple pick-and-place), other tasks might necessitate further fine-tuning (e.g. cooking an omelette). The robot should be able to interactively request demonstrations to compensate for its shortcomings. We seek to answer which demonstrations should be requested from the user to achieve the best performance, as quickly as possible.

If the agent only needs to perform well in a single task, the fine-tuning process conventionally relies on behavioral cloning (Chen et al., 2021; Reed et al., 2022; Bousmalis et al., 2024) of expert demonstrations. As collecting demonstrations is in general costly, the number of demonstrations required, and thus the expert's effort, should be minimized. However, as each demonstration should solve the same task, the allocation of the expert's effort is straightforward. The multi-task case presents the more nuanced problem of selecting which tasks to demonstrate, and when. This motivates the main

focus of this work: *provided a pre-trained policy, how can we maximize multi-task performance with a minimal number of additional demonstrations?*

To address this problem, we propose AMF (*Active Multi-task Fine-tuning*), which selects maximally informative demonstrations. AMF parallels recent work on active supervised fine-tuning of neural networks (Hübotter et al., 2024b). To this end, AMF relies on estimates of the demonstrations' information gain about the expert policy. We prove that in sufficiently regular Markov decision processes, AMF converges to the expert policy. We then focus on practical scenarios where policies are represented as neural networks. We show that, despite additional challenges, AMF can effectively guide active task selection in such settings, leading to better policies after fewer demonstrations. Our contributions are:

- We propose AMF, an algorithm for multi-task policy fine-tuning that maximizes the information gain of demonstrations about the expert policy.
- We prove statistical guarantees for AMF, which extend the results of Hübotter et al. (2024b) to dynamical systems.
- We empirically scale AMF to high-dimensional tasks involving pre-trained neural policies.

## 2 RELATED WORK

Learning-based control and active data selection are both well-established research directions. This section discusses some of the most topical works in either direction, and clarifies the novelty and placement of this work with respect to them.

**Behavioral Cloning**   Numerous imitation learning approaches have been developed with the goal of distilling knowledge from high-quality demonstrations to a control policy (Osa et al., 2018). Within this family of techniques, behavioral cloning (BC, Bain & Sammut, 1995; Ross & Bagnell, 2010) aims to maximize policy performance by minimizing the distance of its actions to demonstrated actions, simply through supervised learning. While BC may suffer from accumulating errors (Ross et al., 2011), its empirical effectiveness has seen increasing support when high-quality demonstrations are readily available (Kumar et al., 2022). Next to recent empirical successes (Chi et al., 2023), formal analysis has also advanced (Spencer et al., 2021; Block et al., 2024a; Belkhale et al., 2024; Foster et al., 2024), and established provable performance guarantees for BC policies (Xu et al., 2020; Maran et al., 2023; Block et al., 2024b).

**Multi-task and Generalist Policies**   Traditionally, behavioral cloning has mostly been deployed in a single-task setting. Multi-task learning in sequential decision-making has largely been investigated in the context of reinforcement learning (Teh et al., 2017; Sodhani et al., 2021; Yu et al., 2021; Sun et al., 2022; Cho et al., 2022; Hendawy et al., 2023). Moreover, the recent rise of multi-task generative models (Brown et al., 2020) has been mirrored by exploration of multi-task, or *generalist* policies, often trained via imitation learning (Reed et al., 2022; Bousmalis et al., 2024; Collaboration, 2023). These recent works mostly build upon algorithms developed for the single-task case, and simply integrate task-conditioning as part of the state. While several works hand-select parts of large, open-source robotics datasets for pre-training (Octo Model Team et al., 2024), active data selection for multi-task fine-tuning has not been addressed. Prior work on meta-learning has studied how one can explicitly meta-learn the ability to adapt to task demonstrations (Finn et al., 2017). We find this capability to emerge even from models that are not explicitly trained in this way, and focus on which demonstrations to obtain.

**Data Selection**   The idea of directing a sampling process to gather information has been central to machine learning research and studied extensively in experimental design (Chaloner & Verdinelli, 1995) and active learning (Settles, 2009). Most work on active data selection summarizes data without focusing on a particular task (e.g., Sener & Savarese, 2017; Ash et al., 2020; Holzmüller et al., 2023; Lightman et al., 2023), which has been predominantly applied to pre-training. Recently, adapting models after pre-training and during deployment has gained interest. Several works, mostly in computer vision, focus on unsupervised fine-tuning on a test instance (Jain & Learned-Miller, 2011; Krause et al., 2018; Sun et al., 2020; Wang et al., 2021b; Chen et al., 2022). We focus instead on *supervised fine-tuning* of learning-based controllers *in dynamical systems*. This necessitates automatic data selection, for which practical methods currently rely on uniform

sampling or externally provided heuristics. Our approach extends work on task-directed data selection (Kothawade et al., 2020; Wang et al., 2021a; Kothawade et al., 2022; Bickford Smith et al., 2023), which has recently been applied to the supervised fine-tuning of large-scale neural networks in vision (Hübotter et al., 2024b) and language (Xia et al., 2024; Hübotter et al., 2024a).

# 3 BACKGROUND

## 3.1 MULTI-TASK REINFORCEMENT LEARNING

The multi-task setting can be modeled by casting the environment as a contextual Markov decision process (MDP) $\mathcal{M} = (\mathcal{S}, \mathcal{A}, \mathcal{C}, P, R, \gamma, \mu_0)$ where $\mathcal{S} \in \mathbb{R}^{N_S}$ and $\mathcal{A} \in \mathbb{R}^{N_{\mathcal{A}}}$ are possibly continuous state and action spaces. $\mathcal{C}$ is a (potentially infinite) set of tasks, with each task represented by an $N_{\mathcal{C}}$-dimensional vector $c \in \mathbb{R}^{N_C}$. $P : \mathcal{S} \times \mathcal{A} \to \Delta(\mathcal{S})$ models the transition probabilities ($\Delta(\mathcal{S})$ represents the set of probability distributions over $\mathcal{S}$), $R : \mathcal{S} \times \mathcal{C} \to \mathbb{R}$ is a scalar reward function, $\gamma \in (0, 1)$ is a discount factor and $\mu_0 \in \Delta(\mathcal{S})$ is the initial state distribution. In this setting, a policy is simply a state-and-task-conditional action distribution $\boldsymbol{\pi} : \mathcal{S} \times \mathcal{C} \to \Delta(\mathcal{A})$[1]. Any given policy induces a task-conditional distribution over trajectories:

$$\boldsymbol{\tau_\pi}\Big((s_0, a_0, s_1, a_1, \dots) \mid c\Big) = \mu_0(s_0) \prod_{t=0}^{\infty} \boldsymbol{\pi}(a_t \mid s_t, c) \cdot P(s_{t+1} \mid s_t, a_t).$$

The discounted returns for a specific task $c \in \mathcal{C}$ or a task distribution $\mu_c \in \Delta(\mathcal{C})$ are, respectively,

$$J_c^{\boldsymbol{\pi}} = \mathop{\mathbb{E}}_{(s_0, \dots) \sim \boldsymbol{\tau_\pi}(c)} \sum_{t=0}^{\infty} \gamma^t R(s_t, c) \quad \text{and} \quad J_{\mu_c}^{\boldsymbol{\pi}} = \mathop{\mathbb{E}}_{c \sim \mu_c} J_c^{\boldsymbol{\pi}}.$$

Reinforcement learning algorithms traditionally aim directly at maximizing $J_{\mu_c}^{\boldsymbol{\pi}}$, which is notoriously challenging. In the scope of this work, we instead consider an imitation learning setting, in which expert demonstrations from an optimal policy $\pi^\star$ are provided. In particular, we focus on behavioral cloning algorithms, which reduce the control problem to a supervised learning problem. Given a set of $N$ task-conditioned, $H$-length trajectories $\hat{\tau}_{1:N} = (s_0^i, a_0^i, \dots, s_{H-1}^i, a_{H-1}^i)_{i=1}^N$ with task labels $c_{1:N}$, behavioral cloning proposes a proxy objective for the policy $\pi$: an empirical estimate of the log-likelihood under the data distribution: $J_{\text{proxy}}^{\boldsymbol{\pi}} = \frac{1}{N} \sum_{i=1}^N \sum_{t=0}^{H-1} \log \boldsymbol{\pi}(a_t^i \mid s_t^i, c_i)$. If trajectories $\tau_{1:N}$ are obtained from the optimal policy, cover the support of the desired task distribution $\mu_c$, and the searched policy class is sufficiently rich, the maximizer of $J_{\text{proxy}}^{\boldsymbol{\pi}}$ will also maximize $J_{\mu_c}^{\boldsymbol{\pi}}$ as $N$ and $H$ increase. However, in general, there is a clear mismatch between $J_{\mu_c}^{\boldsymbol{\pi}}$ and $J_{\text{proxy}}^{\boldsymbol{\pi}}$ (Xu et al., 2020; Maran et al., 2023). Nonetheless, the optimization of $J_{\text{proxy}}^{\boldsymbol{\pi}}$ is a relatively straightforward supervised learning problem, while the full RL problem raises several convergence issues, particularly in the offline setting (Levine et al., 2020). Thus, we use $J_{\mu_c}^{\boldsymbol{\pi}}$ only for evaluation, and carry out optimization through the proxy objective.

## 3.2 ACTIVE POLICY FINE-TUNING

In this work, we consider an active fine-tuning scheme for multi-task policies. The goal is to fine-tune a pre-trained policy to perform well on a desired task distribution $\mu_c$ using as few expert demonstrations as possible. The agent is allowed $N$ sequential queries for demonstrations according to the fine-tuning budget. The $n$-th query should consist of a task $c_n \in \mathcal{C}$. Once the agent selects a task, feedback is received from the optimal policy $\pi^\star : \mathcal{S} \times \mathcal{C} \to \mathcal{A}$ (i.e., an optimal demonstrator). At each round the agent receives an $H$-step demonstration conditioned on the chosen task $c_n$. This can be seen as a single measurement from a stochastic process over trajectories $\boldsymbol{\tau} : \mathcal{C} \to \Delta((\mathcal{S} \times \mathcal{A})^H)$.

Each observed trajectory up to round $n$ is stored in a dataset $(c_{1:n}, \hat{\tau}_{1:n})$, which can be used to fine-tune the policy, and condition the agent's query at step $n+1$. The process is repeated for $N$ rounds, with the goal of producing a fine-tuned policy that maximizes the expected returns for the desired task distribution $\mu_c$.

---

[1]We use $\boldsymbol{\pi}$ and $\pi$ to denote stochastic and deterministic policies, respectively, and $\pi(s, c)$ for realizations.

**Modeling assumptions** We take a Bayesian perspective on active multi-task fine-tuning, by assuming a Bayesian model $\pi$ over policies. We assume that demonstrations follow a noisy expert: $\tilde{\pi}(s, c) = \pi^\star(s, c) + \epsilon(s, c)$ where $\epsilon(s, c)$ is independent noise. We remark, however, that AMF can also be understood from a non-Bayesian perspective as selecting tasks that most quickly minimize the size of frequentist confidence sets around the optimal policy.

## 4 METHOD

The active multi-task fine-tuning problem outlined so far requires active data selection for sample-efficient learning. We thus build on top of principled active learning approaches for non-sequential domains (Hübotter et al., 2024b), and propose AMF, which selects queries that maximize the expected information gain about the expert policy over its occupancy:

$$c_n = \arg\max_{c' \in \mathcal{C}} \mathbb{E}_{\substack{\tau_{1:n-1} \sim \tau(c_{1:n-1}) \\ c \sim \mu_c, \, (s_0, \dots) \sim \tau(c)}} \sum_{t=0}^{H-1} \mathcal{I}(\pi(s_t, c); \tau(c') \mid c_{1:n-1}, \tau_{1:n-1}). \tag{1}$$

We show in Section 4.1 that, under certain regularity assumptions, the policy learned by AMF converges to the expert policy and matches its performance. These results constitute a first-of-its-kind performance guarantee for active multi-task fine-tuning. The main novelty of this guarantee is the extension of prior work to sequential domains where the visited trajectory $(s_0, a_0, s_1, \dots) \sim \tau(c_n)$ is *unknown* when selecting the task $c_n$ for a demonstration. In Section 4.2, we discuss the design choices that make AMF amenable to optimization in practical settings.

### 4.1 PERFORMANCE GUARANTEES

We begin by presenting the performance guarantees for AMF. Our proof builds upon rates for uncertainty reduction, then ties these to probabilistic convergence guarantees to $\pi^\star$, finally resulting in performance guarantees within the MDP. We summarize the main result here, and include a formal proof in Appendix A.

**Informal Assumption 1.** *We make the following assumptions:*

1. *The expert policy $\pi^\star$ is deterministic, Lipschitz-smooth, lies in the reproducing kernel Hilbert space $\mathcal{H}_k(\mathcal{S} \times \mathcal{C})$ of the kernel $k$ with norm $\|\pi^\star\|_k < \infty$ and induces a Lipschitz-smooth Q-function.*

2. *The noise $\epsilon(s, c)$ affecting demonstrations is conditionally $\rho$-sub-Gaussian and bounded.*

3. *The dynamics of the contextual MDP $\mathcal{M}$ are Lipschitz-smooth with bounded support, the initial state distribution $\mu_0$ has bounded support, and the reward is Lipschitz-smooth.*

Under these assumptions, we prove the following performance guarantee for active multi-task behavioral cloning.

**Informal Theorem 2** (Performance guarantees for active multi-task BC). *Let all regularity assumptions hold. If each demonstrated task of length $H$ is selected according to the criterion in Equation 1, then with probability $1 - \delta$ the performance difference between the expert policy $\pi^\star$ and the imitator policy $\pi_n$ after $n$ demonstrations can be upper bounded:*

$$J_{\mu_c}^{\pi^\star} - J_{\mu_c}^{\pi_n} \leq O(\gamma_{(Hn)})/\sqrt{n},$$

*where $\pi_n$ is the mean of $\pi$ at round $n$ and $\gamma_{(Hn)}$ is the maximum information gain about the expert policy from $Hn$ samples, and is sublinear for a large class of problems. The $O(\cdot)$ notation suppresses all multiplicative terms that do not depend on $n$.*

Intuitively, this theorem proves that the imitator will eventually achieve the demonstrator's performance in smooth, regular MDPs with sublinear $\gamma_{(Hn)}$ (for a formal definition, we refer to Lemma 2 in the Appendix). We can also prove a more general result under weaker assumptions: as long as the policy is regular, the imitator will reach the *noisy* expert performance in arbitrary, non-smooth MDPs, albeit only in expectation. A full derivation of this further result can be found in Appendix B.

### 4.2 PRACTICAL ALGORITHMS

Theorem 2 guarantees that, under regularity assumptions, the adaptive demonstration sampling scheme leads to convergence of the imitator's performance to the optimal one. However, this criterion involves state occupancies and a conditional entropy term, which are hard to access or estimate in practice. Thus, here we derive a practical objective to be deployed in general settings. We first rephrase the objective from Equation 1 in its entropy form:

$$
c_n = \arg\min_{c' \in \mathcal{C}} \mathbb{E}_{\substack{\tau_{1:n-1} \sim \boldsymbol{\tau}(c_{1:n-1}),\ \tau' \sim \boldsymbol{\tau}(c') \\ c \sim \mu_c,\ (s_0,\dots) \sim \boldsymbol{\tau}(c)}} \sum_{t=0}^{H-1} \mathcal{H}(\boldsymbol{\pi}(s_t, c) \mid c', \tau', c_{1:n-1}, \tau_{1:n-1}), \tag{2}
$$

where we use the definition of mutual information $\mathcal{I}(\cdot | \boldsymbol{\tau}(c')) = \mathcal{H}(\cdot) - \mathcal{H}(\cdot | \boldsymbol{\tau}(c'))$, drop the first entropy term as it does not depend on $c'$, and rewrite the second entropy term as an expectation over $\boldsymbol{\tau}(c')$. As long as the task space $\mathcal{C}$ is finite and its cardinality is tractable, the $\arg\min$ operator can be evaluated exhaustively, and the expectation over the task distribution $\mu_c$ can be computed exactly. When this is not the case, the $\arg\min$ can be optimized through discretization, or with sampling-based optimizers. The expectation over $\mu_c$ is also not particularly problematic, as it can be computed in closed form (if $\mathcal{C}$ is discrete) or estimated empirically through sampling, as $\mu_c$ is assumed to be known. However, two issues need to be resolved: (i) computing the expectation over the noisy expert's trajectory distribution $\boldsymbol{\tau}$, and (ii) estimating the conditional entropy term $\mathcal{H}(\cdot \mid \cdot)$.

**Occupancy estimation** Computing the expectation over a policy's occupancy over states or trajectories is in general intractable in continuous state spaces. Fortunately, a coarse empirical estimate can be obtained as soon as few expert demonstrations become available. The expectation $\mathbb{E}_{\tau_{1:n-1} \sim \boldsymbol{\tau}(c_{1:n-1})}(\cdot)$ can be estimated through a single sample, which is always available in the form of the trajectories $\hat{\tau}_{1:n-1}$ collected so far, as they have effectively been sampled from $\boldsymbol{\tau}(c_{1:n-1})$. However, the remaining two expectations (i.e., $\mathbb{E}_{\tau' \sim \boldsymbol{\tau}(c')}(\cdot)$ and $\mathbb{E}_{(s_0,\dots) \sim \boldsymbol{\tau}(c)}(\cdot)$) involve the distribution over trajectories for an *arbitrary* task, which might not have been demonstrated yet. However, we observe that, at round $n$, the tasks demonstrated so far induce the empirical distribution $\hat{\mu}_c(\cdot) = \frac{1}{n-1} \sum_{i=1}^{n-1} \delta_{c_i}(\cdot)$, while the trajectories collected similarly induce $\hat{\boldsymbol{\tau}}(\cdot) = \frac{1}{n-1} \sum_{i=1}^{n-1} \delta_{\hat{\tau}_i}(\cdot)$, where $\delta$ indicates the Dirac delta distribution. We can show that expectations over the trajectory distribution for an arbitrary task $c \in \mathcal{C}$ can be estimated through importance sampling (i.e., by sampling trajectories from $\hat{\boldsymbol{\tau}}(\cdot)$ instead of $\boldsymbol{\tau}(\cdot|c)$): $\mathbb{E}_{\tau \sim \boldsymbol{\tau}(\cdot|c)} f(\tau) = \mathbb{E}_{\tau \sim \hat{\boldsymbol{\tau}}(\cdot)} \frac{\boldsymbol{\tau}(\tau|c)}{\hat{\boldsymbol{\tau}}(\tau)} f(\tau)$. The importance weights can then be estimated as

$$
\begin{aligned}
\frac{\boldsymbol{\tau}(\tau|c)}{\hat{\boldsymbol{\tau}}(\tau)} &\approx \frac{\boldsymbol{\tau}(\tau|c)}{\int_{c' \in \mathcal{C}} \hat{\mu}_c(c') \boldsymbol{\tau}(\tau|c')} = \frac{\boldsymbol{\tau}(\tau|c)}{\frac{1}{n-1} \sum_{i=1}^{n-1} \boldsymbol{\tau}(\tau|c_i)} \\
&= \frac{(n-1)\mu_0(s_0) \prod_{t=0}^{H-1} \tilde{\boldsymbol{\pi}}(a_t|s_t, c) P(s_{t+1}|s_t, a_t)}{\sum_{i=1}^{n-1} \mu_0(s_0) \prod_{t=0}^{H-1} \tilde{\boldsymbol{\pi}}(a_t|s_t, c_i) P(s_{t+1}|s_t, a_t)} \\
&= \frac{(n-1) \prod_{t=0}^{H-1} \tilde{\boldsymbol{\pi}}(a_t|s_t, c)}{\sum_{i=1}^{n-1} \prod_{t=0}^{H-1} \tilde{\boldsymbol{\pi}}(a_t|s_t, c_i)} := w(\tau, c)
\end{aligned} \tag{3}
$$

where $\tau = (s_0, a_0, \dots)$ and $\tilde{\boldsymbol{\pi}}$ can be approximated with the current estimate of $\boldsymbol{\pi}$. Intuitively, the likelihood ratio of a trajectory under two different tasks only depends on the likelihood of actions under the policy, and thus does not require knowledge of the MDP. As the estimate may be inaccurate for small numbers of samples, in practice the algorithm can invest the first few rounds to query a single demonstration for each of the tasks (in case they are countable and few) or to sample the task space uniformly. On the other hand, the high-variance of the estimate can be controlled by practical solutions such as clipping. We present the resulting empirical estimate for Equation 2 in full in Appendix C, and a qualitative analysis of importance weights in Appendix J.

**Entropy estimation** The estimation of conditional entropy terms such as $\mathcal{H}(\cdot \mid \cdot)$ has been widely researched in the literature. When the policy is represented through a Gaussian process $GP(\mu, k)$ (Williams & Rasmussen, 2006) with known mean function $\mu$ and kernel $k$,[2] the entropy can be

---

[2] For simplicity, we consider a single-output GP, but generalize to multi-dimensional policies with multi-output GPs in both experiments and formal proofs.

directly quantified by the predicted variance. Let us denote a state-task tuple as $x = (s, c)$, and let $X$ be the sample vector obtained from concatenating states and tasks from previous trajectories (e.g., $c_{1:n-1}, \tau_{1:n-1}, c', \tau'$). The *unconditional* entropy can be measured in closed form as $\mathcal{H}(\boldsymbol{\pi}(x)) = \frac{1}{2}\log(2\pi k(x,x)) + \frac{1}{2}$, and the *conditional* entropy can be obtained by simply replacing the kernel $k$ with $\hat{k}_X(x,x) = k(x,x) - k(x,X)[k(X,X) + \sigma_\epsilon^2 I]^{-1}k(X,x)$, where $\sigma_\epsilon^2$ is the variance of the observation noise $\boldsymbol{\epsilon}(s,c)$, assuming it is distributed according to a zero-mean Gaussian.

Thus, when the policy can be modeled as a GP, the only approximation needed concerns occupancy estimation. We refer to this first, practical instantiation as AMF-GP, and present a general algorithmic framework in Algorithm 1. Application of the method to policies parameterized by neural

---
**Algorithm 1** AMF (practical AMF-NN variant in blue)

---
**Input:** initial policy $\pi_0$, budget $N$, desired task distr. $\mu_c$, batch size $B$
**Output:** fine-tuned policy $\pi_N$
Initialize dataset $\mathcal{D}_0 = \emptyset$; isolate policy parameters
**for** $n \in [0, \ldots, N-1]$ **do**
    Compute $c_n$ as the solution to Eq. 2 (approximated as in Eq. 4)
    Collect new demonstration $\tau_n$ for task $c_n$
    **if** $n + 1 \% B = 0$ **then**
        $\mathcal{D}_{n+1} = \mathcal{D}_{n+1-B} \cup \{c_{n-B+1:n}, \tau_{n-B+1:n}\}$
        Update $\pi_{n+1}$ from $\pi_{n+1-B}$ with $\mathcal{D}_{n+1}$

---

networks will adopt the same scheme. It will also require additional care on three distinct topics: (i) kernel approximations, (ii) batch selection and (iii) forgetting mitigation, as suggested in the Algorithm box, in blue.

**Kernel approximations** When the policy is parameterized through a neural network, estimation of the conditional entropy is far less straightforward. First, we cannot assume the availability of ad-hoc techniques for uncertainty estimation (e.g., Dropout (Srivastava et al., 2014; Gal & Ghahramani, 2016) or ensembles (Lakshminarayanan et al., 2017)), as they might not be featured in pre-trained models. Even if the pre-trained model was perturbed and ensembled for fine-tuning, the ensemble disagreement would not capture the pre-training data distribution. Second, access to pre-training data is in general unrealistic, or hard to manage due to size and ownership of large robotic datasets.

Nevertheless, we can leverage the approximation of neural networks as a linear functions over an embedding space $\boldsymbol{\pi}(s,c;\theta) = \boldsymbol{\beta}^\top \phi_\theta(s,c)$, where both weights $\boldsymbol{\beta}$ and embeddings $\phi_\theta(\cdot)$ exist in a $p$-dimensional latent space (Lee et al., 2019; Khan et al., 2019). This technique does not violate any of the practical constraints listed above, and allows us to adapt the machinery introduced in GP settings. While several embedding strategies exist (Jacot et al., 2018; Devlin et al., 2019; Holzmüller et al., 2023), we adopt loss gradient embeddings (Ash et al., 2020). Assuming the prior $\boldsymbol{\beta} \sim \mathcal{N}(0, I)$, the policy $\boldsymbol{\pi}(s,c;\theta)$ can be modeled by a Gaussian Process with kernel $k_\theta((s,c),(s',c')) = \langle \phi_\theta(s,c), \phi_\theta(s',c') \rangle$. When coupled with this approximation, the conditional entropy objective in Equation 2 can be reformulated:

$$c_n = \arg\max_{c' \in \mathcal{C}} \mathop{\mathbb{E}}_{\substack{\tau_{1:n-1} \sim \boldsymbol{\tau}(c_{1:n-1}),\ \tau' \sim \boldsymbol{\tau}(c') \\ c \sim \mu_c,\ (s_0, \ldots) \sim \boldsymbol{\tau}(c)}} \sum_{t=0}^{H-1} k_\theta((s_t,c),X)[k_\theta(X,X) + \sigma_\epsilon^2 I]^{-1}k_\theta(X,(s_t,c)), \quad (4)$$

where $X$ is vector of states and tasks in $(c', \tau', c_{1:n-1}, \tau_{1:n-1})$. As the collected dataset grows, the conditioning on previous trajectories $\tau_{1:n-1}$ can instead be addressed by fine-tuning the network's parameters $\theta$ (e.g., through conventional gradient descent), resulting in updates in the embedding function $\phi_\theta$.

**Batch selection** Standard practice in *deep* active learning prescribes the collection of *batches* of samples at each round (Gal et al., 2017; Sener & Savarese, 2018; Ducoffe & Precioso, 2018). This necessity is partially addressed by the fact that, in our setting, a single demonstration contains several samples for training the policy (as many as the length $H$ of the demonstration). Nevertheless, we can further leverage the GP approximation to select multiple demonstrations at each round. In fact, a simple recursive greedy selection can provide a constant factor approximation on the information gain objective (Krause & Golovin, 2014; Hübotter et al., 2024b). In practice, when selecting $M$ demonstrations at the $n$-th round, we can select the $m$-th demonstration out of $M$ by applying the criterion in Equation 1, while making sure to condition entropy estimates on both the $m-1$ demonstrations already selected in this round, and the $M \cdot (n-1)$ demonstrations previously collected.

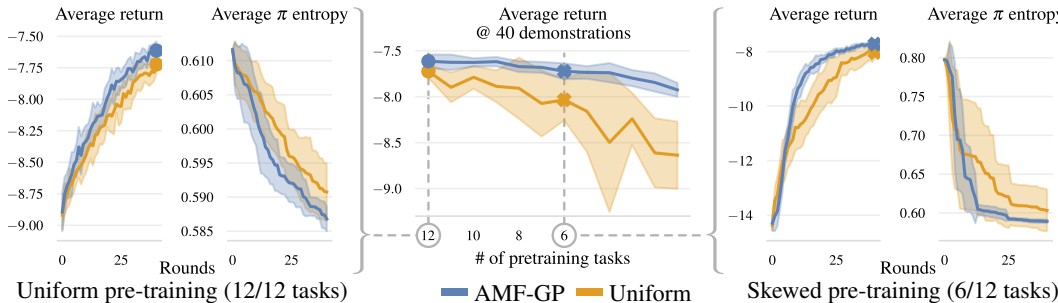

Figure 3: Experiments in GP settings for a 2D integrator (see Figure 2). AMF-GP selects tasks that minimize the policy's posterior entropy and improves the agent's returns faster than uniform task sampling. In the middle, the improvement in final return over the baseline is greater when the pre-training distribution is skewed and includes fewer tasks. We report return and entropy curves for non-skewed and skewed pre-training (left and right, respectively). We report means and 90% simple bootstrap confidence intervals over 10 random seeds ; dots and crosses mark corresponding measurements.

**Dealing with forgetting** A fundamental issue with neural function approximation under shifts to the training distribution is known as *forgetting* (McCloskey & Cohen, 1989; French, 1999), as any further optimization may catastrophically perturb pre-trained parameters. Common strategies for its mitigation often involve rehearsal (Atkinson et al., 2021; Verwimp et al., 2021) or regularization (Kirkpatrick et al., 2017). Unfortunately, the former is not feasible in this setting due to lack of access to pre-training data, and the latter was not found to be empirically effective (see Appendix G). Scale and a diverse pre-training dataset can also mitigate forgetting (Ramasesh et al., 2022), but neither can be controlled during fine-tuning. Thus, we opt for a parameter isolating solution (Rusu et al., 2016; Yu et al., 2020), in which the task-space is partitioned (e.g., uniformly), and a copy of the fine-tuning parameters is stored and trained for each partition, thus avoiding negative interference. Inference can then be performed by selecting the parameter set through a nearest-neighbor lookup in task-space. This solution can be easily applied for limited task sets, and can be scaled to large task spaces through discretization schemes. While parameter isolation prevents constructive interference across tasks, we found it to bring a net benefit during fine-tuning.

By combining the approximations required by AMF-GP with three additional design choices (namely kernel approximation, batch selection and parameter isolation), we obtain a method for active multi-task fine-tuning of policies parameterized via neural networks. We refer to this practical instantiation as AMF-NN.

## 5 EXPERIMENTS

The experiment section is designed to evaluate active multi-task fine-tuning and provide an empirical answer to several questions. We thus reserve a section to each of them.

### 5.1 WHEN IS AMF BENEFICIAL?

When none of the assumptions listed in Section 4.1 is violated, AMF is guaranteed to converge to the optimal policy. We furthermore investigate whether AMF also results in faster empirically faster convergence with respect to naive approaches to data collection. To do so, we compare AMF to uniform i.i.d. sampling from the set of tasks $\mathcal{C}$. First, we consider a classic 2D integrator as a benchmark environment (see Figure 2). The agent is a pointmass initialized in the origin, and can directly control

Figure 2: 2D integrator. Starting from the origin, each task involves reaching a given point on a circle, as shown by differently colored trajectories.

its 2D velocity, which is integrated over the past trajectory to return the current state. We can define an infinite task space, in which each task consists of reaching a point on a circle centered on the origin, and the agent is rewarded with the negative Euclidean distance to it. The evaluation distribution $\mu_c$ assigns equal probability to 12 points in different directions. The initial state distribution is deterministic, dynamics are both deterministic and smooth, while the expert policy is smooth and

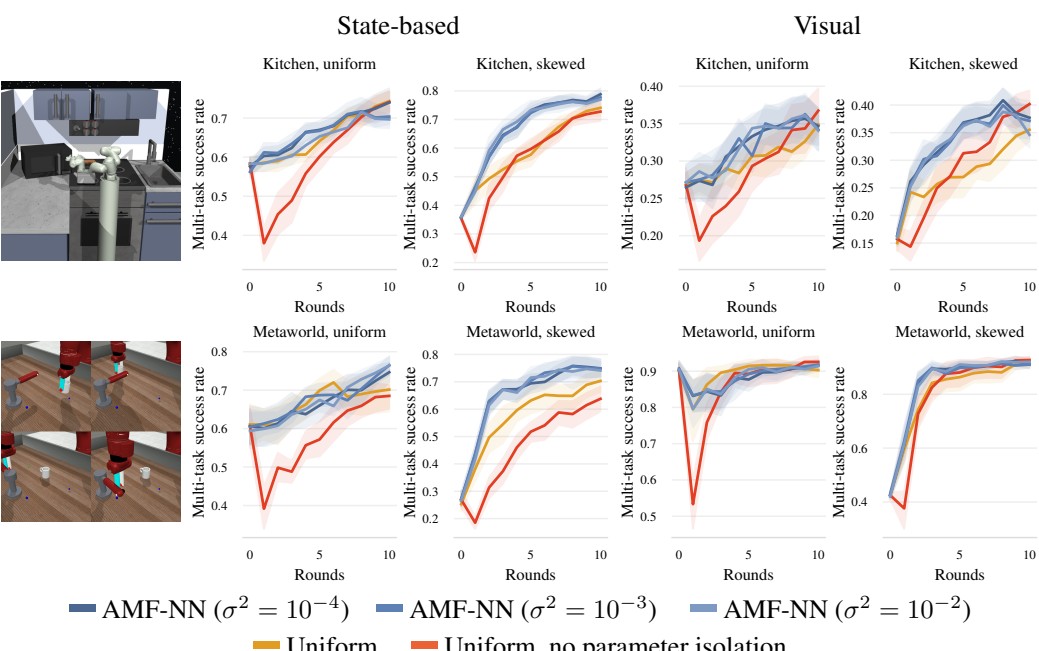

Figure 4: AMF with neural policies in Frankakitchen (top) and Metaworld (bottom). Experiments are repeated for state and RGB inputs (left and right). We evaluate both uniform and skewed pretraining distribution. AMF-NN is overall desirable, and highly beneficial for skewed pre-training distributions. We report means and 90% simple bootstrap confidence intervals over 10 seeds.

corrupted with i.i.d. Gaussian noise. We model the policy as a Gaussian Process with a RBF kernel, and we condition it on a pre-training dataset of 12 noisy demonstrations. We then collect 50 additional demonstrations by running both AMF-GP and uniform sampling.

We first consider a perfectly uniform pre-training regime, in which each evaluation task is demonstrated exactly once (Figure 3, left). As the policy's entropy is minimized, AMF-GP increases the policy's returns at a higher rate compared to uniform sampling of demonstrations. We then extend this evaluation to several pre-training distributions (Figure 3, middle), and compare the final performance of the two methods as the pre-training budget is allocated to a decreasing number of tasks. As the pre-training distribution becomes more skewed (e.g., when only 6/12 tasks are demonstrated in Figure 3, left), we observe that the performance gap between uniform task sampling and AMF-GP grows larger. This is to be expected, as in this case the information gain from the next demonstration heavily depends on the queried task, and taking the $\arg\max$ of the criterion in Equation 1 is significantly better than choosing a random task. Intuitively, in this case, uniform sampling of tasks fails to reliably provide demonstrations for tasks that were observed less often during pre-training.

## 5.2 CAN AMF SCALE TO HIGH-DIMENSIONAL TASKS?

In realistic settings, the assumptions enabling a formal analysis of AMF are soon violated. As the complexity of the environments of interest increases, most modern behavior cloning applications rely on neural networks for policy parameterization (Reed et al., 2022; Chi et al., 2023). Motivated by this pattern, we now study a second version of our method, AMF-NN, and evaluate its ability to scale to complex, high-dimensional tasks. We consider two common benchmarks for multi-task learning, both with a finite set of tasks.

- In Metaworld (Yu et al., 2020) we create a scene with a robotic arm, a cup and a faucet, defining 4 tasks: moving the cup to two distinct positions, opening and closing the faucet.
- In FrankaKitchen (Fu et al., 2020), we consider 5 tasks, namely turning a knob on or off, opening a pivoting or a sliding cabinet, or opening the microwave door.

In both environments, we evaluate AMF-NN when learning from state measurements, as well as from raw pixels. In the first case, the policy is simply parameterized through a MLP, while in the second the MLP receives the embedding of a pre-trained visual encoder (Nair et al., 2022b). The pol-

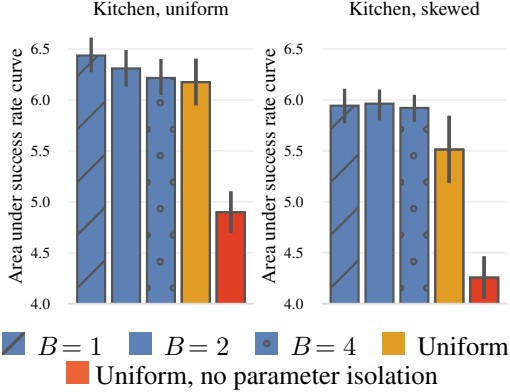

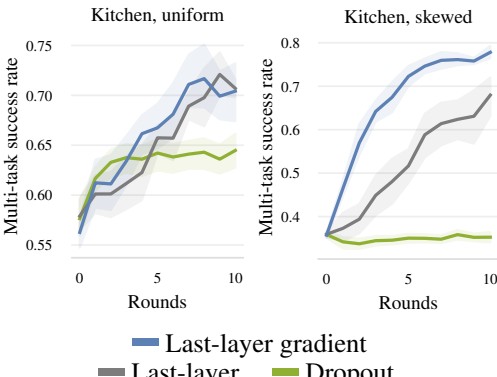

Figure 5: Influence of batch size $B$ over area under success rate curve with 20 demonstrations.

Figure 6: AMF performance with alternative uncertainty estimation schemes.

icy is pre-trained on $\approx 15$ total demonstrations, which we allocate either uniformly across all tasks, or only on half of them, reproducing the uniform and skewed regimes from the previous experiments. Afterwards, we apply AMF-NN for 10 iterations, collecting 2 demonstrations at each iteration.

Figure 4 reports average multi-task success rates at each iteration compared to a random uniform task selection scheme. For the baseline, we additionally report performance without parameter isolation, highlighting performance degradation due to negative gradient interference across tasks. We observe that AMF's performance is on par or better than a uniform task sampling scheme when the prior was trained uniformly on all tasks. However, as reported in the previous section, AMF is very beneficial when the pre-training dataset does not uniformly cover the evaluation tasks. These trends are consistent across both environments, and both modalities. For a qualitative analysis of the strategy induced by AMF, we refer to Appendix H and I.

Finally, Figure 4 also highlights the effect of the only hyperparameter introduced by AMF-NN, namely the noise parameter $\sigma^2$ in the GP approximation. While selecting an excessively high noise level results in slightly greedier behavior and premature convergence, we overall observe that the method is not particularly sensitive to this hyperparameter, performing fairly reliably across 2 orders of magnitude.

### 5.3 IS BATCH-WISE TASK SELECTION IMPORTANT FOR AMF?

In the GP setting, policy fine-tuning and conditioning for batch-wise selection correspond. Thus, it is not necessary to collect batches of demonstrations. This is not the case for AMF-NN. On one hand, collecting batches of training data is computationally beneficial since training can be parallelized effectively (Sener & Savarese, 2018). On the other hand, each query already returns several samples in our setting, which could make batch selection unnecessary. We thus set out to empirically validate which batch size is desirable for active data collection in our setting. While keeping the total budget fixed to 20 demonstrations, we evaluate AMF-NN with batch sizes $B$ spanning from 1 to 4, and report the area under the success rate curve in Figure 5. For convenience, we report the same metric for the uniform selection baselines from Figure 4, for which the data selection strategy does not depend on the batch size. We find that larger batch sizes are not necessarily desirable.

### 5.4 HOW DO UNCERTAINTY ESTIMATES FOR AMF COMPARE?

As entropy estimation is at the core of AMF-NN, we additionally compare the adopted GP approximation with loss-gradient embeddings to other approaches from the literature. In praticular, we also consider an alternative GP approximation using last-layer embeddings (Holzmüller et al., 2023), as well as test-time Dropout (Loquercio et al., 2020). For the latter, each batch is simply filled with demonstrations from the task maximizing *prior* entropy, that is $\arg\max_{c \in \mathcal{C}} \mathbb{E}_{\tau_{1:n-1} \sim \boldsymbol{\tau}(c_{1:n-1}),\, (s_0,...) \sim \boldsymbol{\tau}(c)} \sum_{t=0}^{H-1} \mathcal{H}(\boldsymbol{\pi}(s_t, c) \mid \tau_{1:n-1})$. Both of these schemes are in practice desirable, as they do not require access to action labels. However, we observe that these two schemes are prone to early convergence to suboptimal task choices, or are less effective in driving task selection. Hence, as shown in Figure 6, multi-task performance is in general lower, suggesting that the entropy estimation technique is crucial to AMF-NN.

## 5.5 Can AMF be applied to off-the-shelf models?

As AMF-NN has minimal requirements (essentially, access to a differentiable pre-trained prior is sufficient), it should be widely applicable. In this section, we investigate scaling our evaluation to recently published open-source generalist policies. For this purpose, we chose Octo (Octo Model Team et al., 2024). This model relies on a transformer backbone for integrating multimodal information (in the form of state sensors, camera images and text or RGB task descriptions), and uses a diffusion-based policy head for action prediction (Chi et al., 2023). For computational reasons, we will focus on fine-tuning the action head alone. Octo is pre-trained on a large-scale real-world

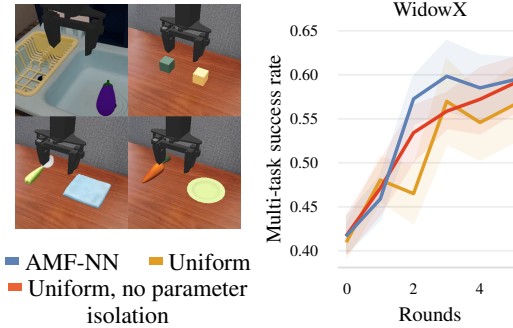

Figure 7: Evaluation on life-like WidowX tasks. AMF-NN can be applied to large-scale settings.

robotic dataset (Collaboration, 2023), and is thus designed for inference on physical hardware. Nonetheless, a recently proposed evaluation suite enables simulated evaluations that statistically correlate with real-world results (Li et al., 2024). We thus collect rollouts from a pre-trained Octo on the WidowX tasks, and filter them to only include successes, akin to self-distillation schemes Bousmalis et al. (2024). On availability of such self-supervised demonstrations, we then apply AMF-NN for 4 iterations, providing 4 demonstrations in each round. The results are reported in Figure 7.

As all evaluation tasks are largely demonstrated in the pre-training dataset (Collaboration, 2023), we find that AMF-NN does not improve significantly upon uniform task collection, confirming the trend we observed for uniform pre-training distributions in Figure 4. Nonetheless, we observe that it constitutes an effective method for data selection, and can be applied as a drop-in replacement for fine-tuning of off-the-shelf models.[3]

## 6 Discussion

As generalist robotic policies gain prominence, a new set of challenges and opportunities emerge. This work responds to this trend by investigating an active multi-task fine-tuning scheme, which adaptively selects the task to be demonstrated for sample-efficient multi-task behavioral cloning. This approach is developed from first principles, extending a formally-motivated, information-based criterion to trajectories over dynamic systems. The resulting method is both formally supported by novel performance guarantees and widely applicable. Moreover, a practical instantiation enables sample-efficient multi-task fine-tuning across GP and neural network policy classes.

Naturally, active multi-task fine-tuning has several limitations. When coupled with neural networks, the algorithm relies on uncertainty estimation techniques, which remain an open problem. While the approximation we leverage is informative in our experiments, AMF could benefit if large pre-trained policies would allow other off-the-shelf uncertainty quantification techniques (e.g., through model ensembling during pre-training). Second, we found the performance of AMF to depend naturally on the pre-training data distribution. While AMF induces efficient learning for skewed pre-training distributions, it naturally brings more modest gains when the pre-trained policy is equally capable for all tasks, and uniform task sampling is sufficient.

On top of addressing the current limitations, this work suggests multiple interesting directions. An extensive empirical evaluation of active fine-tuning with large-scale generalist policies is clearly desirable, but remains infeasible at the moment due to the scarce availability of open-source models and benchmarks. Another future research direction would involve direct estimation of the RL objective, thus removing the dependence on non-equivalent BC proxy objectives.

---

[3]This evaluation also reports an interesting trend, that is a vast reduction in catastrophic forgetting, to the point that parameter isolation is not necessary. This anecdotal evidence can be seen as an instance of a general trend of mitigated catastrophic forgetting in large models (Ramasesh et al., 2022).

REPRODUCIBILITY STATEMENT

Appendix L describes implementation details and hyperparameters across all experiments. We additionally open-source a clean and commented implementation of AMF-GP and AMF-NN at the anonymous website.

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

## A  PERFORMANCE GUARANTEES UNDER REGULARITY ASSUMPTIONS

This sections retrieves guarantees on the performance of the imitator policy as a function of the number of provided demonstrations $n$. At first, this analysis focuses on policies over a single-dimensional action space. An extension to multi-dimensional outputs is introduced later on. The general sketch of the proof can be informally described as follows:

- we first introduce the regularity assumptions required for the guarantees;

- we then show that, in Lipschitz, bounded MDPs, the effect of stochasticity on information gain at each round can be controlled;

- we show how the variance over the imitator's policy shrinks according to the maximum information gain at each round, which in turn depends on the maximum information gain over a set of queries to the expert;

- starting from the previous result, we leverage a well-known theorem (Abbasi-Yadkori, 2013) to retrieve a probabilistic, anytime guarantee on the error of the imitator;

- we quantify the relationship between the imitator's error and its performance, thus retrieving our main theoretical result.

### A.1  ASSUMPTIONS

It is clear that bounding imitation performance would be hopeless without any regularity assumption, as slight errors in the imitator's policy could result in arbitrary differences in return. We thus introduce the following:

**Assumption 1.** (Regular, noisy policy) *We assume that the optimal policy $\pi^\star \sim GP(\mu, k)$ with known mean function $\mu$ and kernel $k$. Furthermore the noise $\epsilon(s, c)$ is mutually independent and zero-mean Gaussian, with known variance $\rho^2(s, c) > 0$ for all $(s, c) \in \mathcal{S} \times \mathcal{C}$.*

In order to motivate further assumptions, let us recall the criterion from Equation 1:

$$c_n = \arg\max_{c' \in \mathcal{C}} \mathbb{E}_{\substack{\tau_{1:n-1} \sim \boldsymbol{\tau}(c_{1:n-1}) \\ c \sim \mu_c, (s_0, \dots) \sim \boldsymbol{\tau}(c)}} \sum_{t=0}^{H-1} \mathcal{I}(\boldsymbol{\pi}(s_t, c); \boldsymbol{\tau}(c') \mid c_{1:n-1}, \tau_{1:n-1}). \tag{5}$$

Through this section, we will use a slightly more precise formulation:

$$c_n = \arg\max_{c' \in \mathcal{C}} \mathbb{E}_{\substack{\tau_{1:n-1} \sim \boldsymbol{\tau}(c_{1:n-1}), \tau \sim \boldsymbol{\tau}(c') \\ c \sim \mu_c, (s_0, \dots) \sim \boldsymbol{\tau}(c)}} \sum_{t=0}^{H-1} \mathcal{I}(\boldsymbol{\pi}(s_t, c); \tilde{\boldsymbol{\pi}}(\tau, c') \mid c_{1:n-1}, \tau_{1:n-1}), \tag{6}$$

in which we clarify that the mutual information is only computed with respect to the actions of the noisy expert, and overload the notation with $\tilde{\boldsymbol{\pi}}(\tau, c') = (\tilde{\boldsymbol{\pi}}(s_i, c'))_0^{H-1}$ for $\tau = (s_i, a_i)_0^{H-1}$. The criterion selects the task $c_n$ with the greatest expected mutual information between the policy and the trajectory associated with the task. We note that, the objective produces a fully deterministic sequence of tasks, as all stochasticity is resolved in the expectation. Nevertheless, the actual sequence of states at which the demonstrator is queried remains stochastic. For this reason, we require the following two sets of assumptions to ensure that information gained along empirical trajectories is not arbitrarily smaller than the expected one.

**Assumption 2.** (Lipschitz, bounded MDP and policy) *Given the contextual MDP $\mathcal{M} = (\mathcal{S}, \mathcal{A}, \mathcal{C}, P, R, \gamma, \mu_0)$ and the noisy expert $\tilde{\boldsymbol{\pi}}$ we assume that, for every $\{(s, c, a), (s', c', a')\} \subseteq \mathcal{S} \times \mathcal{C} \times \mathcal{A}$:*

- *the support of the initial state distribution $\mu_0$ is bounded by an $\epsilon_{\mu_0}$-ball*

$$\max_{s_l, s_h \in \text{supp}(\mu_0)} \|s_h - s_l\|_2 \le \epsilon_{\mu_0},$$

- *the transition kernel $P$ is $L_P$-smooth*

$$\mathcal{W}(P(\cdot|s, a), P(\cdot|s', a')) \le L_P \cdot d((s, a), (s', a')),$$

*where $d((s, a), (s', a')) = \|s - s'\|_2 + \|a - a'\|_2$, and $\mathcal{W}(\cdot, \cdot)$ is the Wasserstein 1-distance with respect to $d(\cdot, \cdot)$; furthermore, the support of $P(\cdot|s, a)$ is bounded by an $\epsilon_P$-ball*

$$\max_{s_l, s_h \in \mathrm{supp}(P(\cdot|s,a))} \|s_h - s_l\|_2 \leq \epsilon_P,$$

- *the reward function $R$ is $L_R$-smooth*

$$|R(s, c, a) - R(s', c', a')| < L_R \cdot d((s, c, a), (s', c', a')),$$

*where $d((s, c, a), (s', c', a')) = \|s - s\|_2 + \|c - c'\|_2 + \|a - a'\|_2$,*

- *the noisy expert $\tilde{\boldsymbol{\pi}}$ is $L_\pi$-smooth*

$$\mathcal{W}(\tilde{\boldsymbol{\pi}}(\cdot|s, c, a), P(\cdot|s', c', a')) \leq L_\pi \cdot d((s, c, a), (s', c', a')),$$

*where $d((s, c, a), (s', c', a')) = \|s - s\|_2 + \|c - c'\|_2 + \|a - a'\|_2$ and $\mathcal{W}(\cdot, \cdot)$ is the 1-Wasserstein distance with respect to $d(\cdot, \cdot)$; furthermore, the support of $\tilde{\boldsymbol{\pi}}(\cdot|s, a)$ is bounded by an $\epsilon_\pi$-ball*

$$\max_{a_l, a_h \in \mathrm{supp}(\tilde{\boldsymbol{\pi}}(\cdot|s,c))} \|a_h - a_l\|_2 \leq \epsilon_P,$$

- *finally, the Q-function for the expert $\pi^\star$ is $L_Q$-smooth*

$$|Q^{\pi^\star}(s, c, a) - Q^{\pi^\star}(s', c', a')| \leq L_Q \cdot d((s, c, a), (s', c', a')).$$

We note that smoothness of the noisy expert is guaranteed by construction if the expert $\pi^\star$ is $L_\pi$-smooth.

**Assumption 3.** (Smooth MI) *For every pair of sequences of trajectories $\{\tau_{1:n-1}, \tau'_{1:n-1}\} \subseteq (\mathcal{S} \times \mathcal{A})^{H(n-1)}$, $(s, c) \in \mathcal{S} \times \mathcal{C}$, $c_{1:n-1} \in \mathcal{C}^{n-1}$, $\tilde{\tau} \in (\mathcal{S} \times \mathcal{A})^H$ and $c_n \in \mathcal{C}$, we assume that the mutual information at step $n$ is $L_I$-smooth with respect to the mean square deviation of collected trajectories:*

$$|\mathcal{I}(\boldsymbol{\pi}(s, c); \tilde{\boldsymbol{\pi}}(\tilde{\tau}, c_n)|c_{1:n-1}, \tau_{1:n-1}) - \mathcal{I}(\boldsymbol{\pi}(s, c); \tilde{\boldsymbol{\pi}}(\tilde{\tau}, c_n)|c_{1:n-1}, \tau'_{1:n-1})| \leq L_I \cdot d(\tau_{1:n-1}, \tau'_{1:n-1}),$$

*where $d((s_{0,1}, a_{0,1}, \ldots s_{H,n-1}, a_{H,n-1}), (s'_{0,1}, a'_{0,1}, \ldots s'_{H,n-1}, a'_{H,n-1})) = \frac{1}{n-1} \sum_{m=1}^{n-1} (\sum_{t=0}^{H-1} \|s_{t,m} - s'_{t,m}\|_2^2 + \|a_{t,m} - a'_{t,m}\|_2^2)^{\frac{1}{2}}$ is the mean square deviation over the concatenation of trajectories.*

### A.2 PROOF

We first prove that, under Assumptions 2 and 3, the effect of stochasticity on the mutual information at step $n$ is bounded.

**Lemma 1.** *Let Assumptions 2 and 3 hold. Fix a sequence of tasks $c_{1:n-1}$ and consider two arbitrary sequences of trajectories $\tau_{1:n-1}$ and $\tau'_{1:n-1}$ sampled from $\boldsymbol{\tau}(c_{1:n-1})$. Fix one state-task pair $(s, c) \in \mathcal{S} \times \mathcal{C}$, one task $c_n \in \mathcal{C}$ and one trajectory $\tilde{\tau} \sim \boldsymbol{\tau}(c_n)$. Let $\epsilon_n = 8H^{\frac{3}{2}}(1 + \max(L_P, L_\pi))^H \max(\epsilon_0, \epsilon_\pi, \epsilon_P)$. The difference in mutual information when conditioning on the two sequences of trajectories can be bounded:*

$$|\mathcal{I}(\boldsymbol{\pi}(s, c); \tilde{\boldsymbol{\pi}}(\tilde{\tau}, c_n)|\tau_{i:n-1}) - \mathcal{I}(\boldsymbol{\pi}(s, c); \tilde{\boldsymbol{\pi}}(\tilde{\tau}, c_n)|\tau'_{i:n-1})| \leq \epsilon_n.$$

*Proof.* Under Assumption 3 it is sufficient to show that stochasticity in the MDP does not cause the demonstrator's trajectories to deviate excessively. This is a direct consequence of smoothness and boundedness, which we assume in Assumption 2, and can be shown by induction. Let us fix a task $c_n \in \mathcal{C}$ and consider two trajectories $\tau, \tau' \sim \boldsymbol{\tau}(c_n)$. For the two initial states $(s_0, s'_0)$, boundedness of the initial state distribution $\mu_0$ implies that $\|s_0 - s'_0\|_2 \leq \epsilon_{\mu_0}$. Now, assuming that the distance

between two states $(s_t, s'_t)$ is bounded as $\|s_t - s'_t\|_2 \le \epsilon_t$, we have that

$$\epsilon_{t+1} := \|s_{t+1} - s'_{t+1}\|_2 \tag{7}$$

$$\overset{(i)}{\le} \mathcal{W}(P(\cdot|s_t, a_t), P(\cdot|s'_t, a'_t)) + 2\epsilon_P \tag{8}$$

$$\le L_P \cdot (\|s_t - s'_t\|_2 + \|a_t - a'_t\|_2) + 2\epsilon_P \tag{9}$$

$$= L_P \cdot (\epsilon_t + \|a_t - a'_t\|_2) + 2\epsilon_P \tag{10}$$

$$\overset{(ii)}{\le} L_P \cdot (\epsilon_t + \mathcal{W}(\tilde{\boldsymbol{\pi}}(\cdot|s_t, c_t), \tilde{\boldsymbol{\pi}}(\cdot|s'_t, c'_t)) + 2\epsilon_\pi) + 2\epsilon_P \tag{11}$$

$$\le L_\pi \cdot (\epsilon_t + L_\pi \cdot \|s_t - s'_t\|_2 + 2\epsilon_\pi) + 2\epsilon_P \tag{12}$$

$$= L_P \cdot (\epsilon_t + L_\pi \cdot \epsilon_t + 2\epsilon_\pi) + 2\epsilon_P \tag{13}$$

$$= L_P \cdot ((L_\pi + 1) \cdot \epsilon_t + 2\epsilon_\pi) + 2\epsilon_P \tag{14}$$

$$= L_P(1 + L_\pi)\epsilon_t + 2(L_P\epsilon_\pi + \epsilon_P) \tag{15}$$

$$:= A\epsilon_t + B, \tag{16}$$

where Lemma 9 was used in (i) and (ii); Assumption 2 and the fact that $c_t = c'_t$ were used through the rest of the derivation. The recurrence relation can be easily unrolled as

$$\epsilon_t \le A^t \epsilon_0 + \sum_{i=0}^{t-1} A^i B \tag{17}$$

$$\le A^t \epsilon_0 + \max(A^{t-1}, 1)Bt \tag{18}$$

$$\le \max(A, 1)^t (\epsilon_0 + Bt) \tag{19}$$

$$= \max(L_P(1 + L_\pi), 1)^t (\epsilon_0 + 2t(L_P\epsilon_\pi + \epsilon_P)) \tag{20}$$

$$\le (1 + L_P)^t (1 + L_\pi)^t (\epsilon_0 + 2t((1 + L_P)\epsilon_\pi + \epsilon_P)) \tag{21}$$

$$\le (1 + L_P)^t (1 + L_\pi)^t (2t(1 + L_P)\max(\epsilon_0, \epsilon_\pi, \epsilon_P)) \tag{22}$$

$$= 2t(1 + L_P)^{t+1}(1 + L_\pi)^t \max(\epsilon_0, \epsilon_\pi, \epsilon_P), \tag{23}$$

thus bounding the L2 distances between states at each step of the trajectory $\epsilon_t = \|s_t - s'_t\|_2$. We note that the distance between actions can also be easily bound by Lemma 9: $\|a_t - a'_t\|_2 \le L_\pi \epsilon_t + 2\epsilon_\pi$. This can in turn be related to distances over trajectories. Let us fix $c_{1:n-1} \in \mathcal{C}$ and consider $\tau_{1:n-1}, \tau'_{1:n-1} \sim \boldsymbol{\tau}(c_{1:n-1})$. We have that

$$d(\tau_{1:n-1}, \tau'_{1:n-1}) = \frac{1}{n-1} \sum_{m=1}^{n-1} (\sum_{t=0}^{H-1} \|s_{t,m} - s'_{t,m}\|_2^2 + \|a_{t,m} - a'_{t,m}\|_2^2)^{\frac{1}{2}} \tag{24}$$

$$\le (\sum_{t=0}^{H-1} \epsilon_t^2 + (L_\pi \epsilon_t + 2\epsilon_\pi)^2)^{\frac{1}{2}} \tag{25}$$

$$\le (\sum_{t=0}^{H-1} \epsilon_t^2 + (L_\pi \epsilon_t + \epsilon_t)^2)^{\frac{1}{2}} \tag{26}$$

$$= (\sum_{t=0}^{H-1} \epsilon_t^2 + (1 + L_\pi)^2 \epsilon_t^2)^{\frac{1}{2}} \tag{27}$$

$$\le (\sum_{t=0}^{H-1} 2(1 + L_\pi)^2 \epsilon_t^2)^{\frac{1}{2}} \tag{28}$$

$$= (2(1 + L_\pi)^2 \sum_{t=0}^{H-1} \epsilon_t^2)^{\frac{1}{2}} \tag{29}$$

$$= \sqrt{2}(1 + L_\pi)(\sum_{t=0}^{H-1} \epsilon_t^2)^{\frac{1}{2}} \tag{30}$$

$$\le \sqrt{2}(1 + L_\pi)(H\epsilon_{H-1}^2)^{\frac{1}{2}} \tag{31}$$

$$= \sqrt{2H}(1 + L_\pi)\epsilon_{H-1} \tag{32}$$

$$\leq \sqrt{2H}(1 + L_\pi) \cdot 2(H-1)(1 + L_P)^H(1 + L_\pi)^{H-1} \max(\epsilon_0, \epsilon_\pi, \epsilon_P) \tag{33}$$

$$\leq 4H^{\frac{3}{2}}(1 + L_P)^H(1 + L_\pi)^H \max(\epsilon_0, \epsilon_\pi, \epsilon_P) \tag{34}$$

$$\leq 8H^{\frac{3}{2}}(1 + \max(L_P, L_\pi))^H \max(\epsilon_0, \epsilon_\pi, \epsilon_P). \tag{35}$$

Having obtained an upper bound on the distance between sequences of trajectories, the result follows naturally from smoothness of mutual information according to Assumption 3.

$\square$

We can now focus on the main result. We start by introducing an important measure, quantifying the maximum information gain at each round:

$$\Gamma_n := \max_{c' \in \mathcal{C}} \psi_n(c') = \max_{c' \in \mathcal{C}} \mathbb{E}_{\substack{\tau_{1:n-1} \sim \boldsymbol{\tau}(c_{1:n-1}) \\ \tau' \sim \boldsymbol{\tau}(c') \\ c \sim \mu_c, (s_0, \dots) \sim \boldsymbol{\tau}(c)}} \sum_{t=0}^{H-1} \mathcal{I}(\boldsymbol{\pi}(s_t, c); \tilde{\boldsymbol{\pi}}(\tau', c') \mid c_{1:n-1}, \tau_{1:n-1}) \tag{36}$$

We note that the criterion in Equation 6 takes the $\arg\max$ of the same quantity $\Gamma_n$ maximizes over. As common in the literature (Bogunovic et al., 2016; Kothawade et al., 2020; Hübotter et al., 2024b), we make a standard assumption on diminishing informativeness.

**Assumption 4.** *For each $n, i \in \mathbb{N}$ with $i \leq n$, the maximum information gain at round $n$ is not greater than the maximum information gain at round $i$:*

$$\Gamma_n \leq \Gamma_i.$$

This can be leveraged to show that the expected mutual information is sublinear in the number of rounds $n$. From this point, we overload the notation and allow policies (e.g., $\boldsymbol{\pi}$) to map vector to random vectors, that is $\boldsymbol{\pi}((x_0, \dots, x_{n-1})) = (\boldsymbol{\pi}(x_0), \dots, \boldsymbol{\pi}(x_{n-1}))$ for $(x_0, \dots, x_{n-1}) \in (\mathcal{S} \times \mathcal{C})^n$.

**Lemma 2.** *Under Assumptions 1 and 4, if $(c_0, \dots, c_n)$ follows the criterion in Equation 6, then $\Gamma_n \leq \frac{H}{n}\gamma_{(Hn)}$, where $\gamma_{(Hn)} = \max_{\substack{X \subseteq \mathcal{S} \times \mathcal{C} \\ |X| \leq Hn}} \mathcal{I}(\boldsymbol{\pi}(\mathcal{S} \times \mathcal{C}); \tilde{\boldsymbol{\pi}}(X))$.*

*Proof.*

$$\Gamma_n = \frac{1}{n}\sum_{i=0}^{n-1}\Gamma_n \tag{37}$$

$$\overset{(i)}{\leq} \frac{1}{n}\sum_{i=0}^{n-1}\Gamma_i \tag{38}$$

$$= \frac{1}{n}\sum_{i=0}^{n-1} \max_{c' \in \mathcal{C}} \mathbb{E}_{\substack{\tau_{1:n-1} \sim \boldsymbol{\tau}(c_{1:n-1}) \\ \tau' \sim \boldsymbol{\tau}(c') \\ c \sim \mu_c, (s_0, \dots) \sim \boldsymbol{\tau}(c)}} \sum_{t=0}^{H-1} \mathcal{I}(\boldsymbol{\pi}(s_0, c); \tilde{\boldsymbol{\pi}}(\tau', c') \mid c_{1:n-1}, \tau_{1:n-1}) \tag{39}$$

$$\overset{(ii)}{=} \frac{1}{n}\sum_{i=0}^{n-1} \mathbb{E}_{\substack{\tau_{1:n-1} \sim \boldsymbol{\tau}(c_{1:n-1}) \\ \tau' \sim \boldsymbol{\tau}(c_n) \\ c \sim \mu_c, (s_0, \dots) \sim \boldsymbol{\tau}(c)}} \sum_{t=0}^{H-1} \mathcal{I}(\boldsymbol{\pi}(s_0, c); \tilde{\boldsymbol{\pi}}(\tau', c_n) \mid c_{1:n-1}, \tau_{1:n-1}) \tag{40}$$

$$= \frac{1}{n} \mathbb{E}_{\substack{c \sim \mu_c \\ (s_0, \dots) \sim \boldsymbol{\tau}(c)}} \sum_{t=0}^{H-1} \mathbb{E}_{\substack{\tau_n \sim \boldsymbol{\tau}(c_n) \\ \tau_{1:n-1} \sim \boldsymbol{\tau}(c_{1:n-1})}} \sum_{i=0}^{n-1} \mathcal{I}(\boldsymbol{\pi}(s_0, c); \tilde{\boldsymbol{\pi}}(\tau_n, c_n) \mid c_{1:n-1}, \tau_{1:n-1}) \tag{41}$$

$$\overset{(iii)}{=} \frac{1}{n} \mathbb{E}_{\substack{c \sim \mu_c \\ (s_0, \dots) \sim \boldsymbol{\tau}(c)}} \sum_{t=0}^{H-1} \mathbb{E}_{\substack{\tau_n \sim \boldsymbol{\tau}(c_n) \\ \tau_{1:n-1} \sim \boldsymbol{\tau}(c_{1:n-1})}} \mathcal{I}(\boldsymbol{\pi}(s_0, c); \tilde{\boldsymbol{\pi}}(\tau_{1:n}, c_{1:n})) \tag{42}$$

$$\leq \frac{1}{n} \mathop{\mathbb{E}}_{\substack{c \sim \mu_c \\ (s_0,\dots) \sim \boldsymbol{\tau}(c)}} \sum_{t=0}^{H-1} \max_{\substack{X \subseteq \mathcal{S} \times \mathcal{C} \\ |X|=Hn}} \mathcal{I}(\boldsymbol{\pi}(s_0,c); \tilde{\boldsymbol{\pi}}(X)) \tag{43}$$

$$\leq \frac{1}{n} \mathop{\mathbb{E}}_{\substack{c \sim \mu_c \\ (s_0,\dots) \sim \boldsymbol{\tau}(c)}} \sum_{t=0}^{H-1} \max_{\substack{X \subseteq \mathcal{S} \times \mathcal{C} \\ |X|=Hn}} \mathcal{I}(\boldsymbol{\pi}(\mathcal{S} \times \mathcal{C}); \tilde{\boldsymbol{\pi}}(X)) \tag{44}$$

$$= \frac{H}{n} \max_{\substack{X \subseteq \mathcal{S} \times \mathcal{C} \\ |X|=Hn}} \mathcal{I}(\boldsymbol{\pi}(\mathcal{S} \times \mathcal{C}); \tilde{\boldsymbol{\pi}}(X)) \tag{45}$$

$$= \frac{H}{n} \gamma_{(Hn)} \tag{46}$$

where (i) follows from Assumption 4, (ii) follows from Equation 6, (iii) is due to the chain rule of mutual information. We note that $\gamma_n = \max_{X \subseteq \mathcal{S} \times \mathcal{C}, |X| \leq n} \mathcal{I}(\boldsymbol{\pi}(\mathcal{S} \times \mathcal{C}); \tilde{\boldsymbol{\pi}}(X))$ is sublinear for a large class of GPs. In this cases, a looser upper bound would be $H^2 \frac{\gamma_n}{n}$. $\qquad \square$

This bound on expected round-wise mutual information can then be leveraged to describe how the total variance shrinks over rounds.

**Lemma 3.** *(Uniform convergence of marginal variance, following Hübotter et al. (2024b)) Under Assumption 1, 2 and 3, for any $n \geq 0$ and $(s,c) \in \mathcal{S} \times \mathcal{C}$,*

$$\sigma_n^2(s,c) \leq (1 + \epsilon_n) \frac{2\bar{\sigma}^2 \Gamma_n}{\tau_{min}^2},$$

*where $\bar{\sigma}^2 = \max_{(s,c) \in \mathcal{S} \times \mathcal{C}} \sigma_0^2(s,c) + \rho^2(s,c)$ and $\tau_{min} = \min_{s,c \in \mathcal{S} \times \mathcal{C}} \mathbb{E}_{\tau \sim \boldsymbol{\tau}(c)} \mathbf{1}_{s \in \tau}$.*

*Proof.*

$$\sigma_n^2(s,c) = \text{Var}[\boldsymbol{\pi}(s,c) \mid c_{1:n}, \tau_{1:n}] \tag{47}$$

$$= \big(\text{Var}[\boldsymbol{\pi}(s,c) \mid c_{1:n}, \tau_{1:n}] + \rho^2(s,c)\big) - \rho^2(s,c) \tag{48}$$

$$= \text{Var}[\tilde{\boldsymbol{\pi}}(s,c) \mid c_{1:n}, \tau_{1:n}] - \text{Var}[\tilde{\boldsymbol{\pi}}(s,c) \mid \boldsymbol{\pi}(s,c), c_{1:n}, \tau_{1:n}] \tag{49}$$

$$\overset{(i)}{\leq} \bar{\sigma}^2 \log \left( \frac{\text{Var}[\tilde{\boldsymbol{\pi}}(s,c) \mid c_{1:n}, \tau_{1:n}]}{\text{Var}[\tilde{\boldsymbol{\pi}}(s,c) \mid \boldsymbol{\pi}(s,c), c_{1:n}, \tau_{1:n}]} \right) \tag{50}$$

$$= 2\bar{\sigma}^2 \mathcal{I}(\boldsymbol{\pi}(s,c); \tilde{\boldsymbol{\pi}}(s,c) \mid c_{1:n}, \tau_{1:n}) \tag{51}$$

$$= 2\bar{\sigma}^2 \frac{1}{\mathbb{E}_{\tau \sim \boldsymbol{\tau}(c)} \mathbf{1}_{s \in \tau}} \mathop{\mathbb{E}}_{\tau \sim \boldsymbol{\tau}(c)} \mathbf{1}_{s \in \tau} \mathcal{I}(\boldsymbol{\pi}(s,c); \tilde{\boldsymbol{\pi}}(s,c) \mid c_{1:n}, \tau_{1:n}) \tag{52}$$

$$\overset{(ii)}{\leq} \frac{2\bar{\sigma}^2}{\tau_{\min}} \mathop{\mathbb{E}}_{\tau \sim \boldsymbol{\tau}(c)} \mathbf{1}_{s \in \tau} \mathcal{I}(\boldsymbol{\pi}(s,c); \tilde{\boldsymbol{\pi}}(s,c) \mid c_{1:n}, \tau_{1:n}) \tag{53}$$

$$\leq \frac{2\bar{\sigma}^2}{\tau_{\min}} \mathop{\mathbb{E}}_{\tau \sim \boldsymbol{\tau}(c)} \mathbf{1}_{s \in \tau} \mathcal{I}(\boldsymbol{\pi}(s,c); \tilde{\boldsymbol{\pi}}(\tau,c) \mid c_{1:n}, \tau_{1:n}) \tag{54}$$

$$\leq \frac{2\bar{\sigma}^2}{\tau_{\min}} \mathop{\mathbb{E}}_{\tau \sim \boldsymbol{\tau}(c)} \mathcal{I}(\boldsymbol{\pi}(s,c); \tilde{\boldsymbol{\pi}}(\tau,c) \mid c_{1:n}, \tau_{1:n}) \tag{55}$$

$$\leq \frac{2\bar{\sigma}^2}{\tau_{\min}} \frac{1}{\mathbb{E}_{\substack{c \sim \mu_c \\ (s_0,\dots) \sim \boldsymbol{\tau}(c)}} \mathbf{1}_{s \in (s_0,\dots)}} \mathop{\mathbb{E}}_{\substack{c \sim \mu_c \\ (s_0,\dots) \sim \boldsymbol{\tau}(c)}} \mathbf{1}_{s \in (s_0,\dots)} \mathop{\mathbb{E}}_{\tau \sim \boldsymbol{\tau}(c)} \mathcal{I}(\boldsymbol{\pi}(s,c); \tilde{\boldsymbol{\pi}}(\tau,c) \mid c_{1:n}, \tau_{1:n}) \tag{56}$$

$$\leq \frac{2\bar{\sigma}^2}{\tau_{\min}^2} \mathop{\mathbb{E}}_{\substack{c \sim \mu_c \\ (s_0,\dots) \sim \boldsymbol{\tau}(c)}} \mathbf{1}_{s \in (s_0,\dots)} \mathop{\mathbb{E}}_{\tau \sim \boldsymbol{\tau}(c)} \mathcal{I}(\boldsymbol{\pi}(s,c); \tilde{\boldsymbol{\pi}}(\tau,c) \mid c_{1:n}, \tau_{1:n}) \tag{57}$$

$$= \frac{2\bar{\sigma}^2}{\tau_{\min}^2} \mathop{\mathbb{E}}_{\substack{c \sim \mu_c \\ \tau \sim \boldsymbol{\tau}(c) \\ (s_0,\dots) \sim \boldsymbol{\tau}(c)}} \mathbf{1}_{s \in (s_0,\dots)} \mathcal{I}(\boldsymbol{\pi}(s,c); \tilde{\boldsymbol{\pi}}(\tau,c) \mid c_{1:n}, \tau_{1:n}) \tag{58}$$

$$\leq \frac{2\bar{\sigma}^2}{\tau_{\min}^2} \mathop{\mathbb{E}}_{\substack{c \sim \mu_c \\ \tau \sim \boldsymbol{\tau}(c) \\ (s_0,\dots) \sim \boldsymbol{\tau}(c)}} \mathbf{1}_{s \in (s_0,\dots)} \sum_{t=0}^{H-1} \mathcal{I}(\boldsymbol{\pi}(s_t, c); \tilde{\boldsymbol{\pi}}(\tau, c) \mid c_{1:n}, \tau_{1:n}) \tag{59}$$

$$\leq \frac{2\bar{\sigma}^2}{\tau_{\min}^2} \mathop{\mathbb{E}}_{\substack{c \sim \mu_c \\ \tau \sim \boldsymbol{\tau}(c) \\ (s_0,\dots) \sim \boldsymbol{\tau}(c)}} \sum_{t=0}^{H-1} \mathcal{I}(\boldsymbol{\pi}(s_t, c); \tilde{\boldsymbol{\pi}}(\tau, c) \mid c_{1:n}, \tau_{1:n}) \tag{60}$$

$$\stackrel{(iii)}{\leq} (1 + \epsilon_n) \frac{2\bar{\sigma}^2}{\tau_{\min}^2} \mathop{\mathbb{E}}_{\substack{c \sim \mu_c \\ \tau \sim \boldsymbol{\tau}(c) \\ (s_0,\dots) \sim \boldsymbol{\tau}(c)}} \sum_{t=0}^{H-1} \mathop{\mathbb{E}}_{\tau_{1:n-1} \sim \boldsymbol{\tau}(c_{1:n-1})} \mathcal{I}(\boldsymbol{\pi}(s_t, c); \tilde{\boldsymbol{\pi}}(\tau, c) \mid c_{1:n}, \tau_{1:n}) \tag{61}$$

$$= (1 + \epsilon_n) \frac{2\bar{\sigma}^2}{\tau_{\min}^2} \mathop{\mathbb{E}}_{\substack{\tau_{1:n-1} \sim \boldsymbol{\tau}(c_{1:n-1}) \\ \tau \sim \boldsymbol{\tau}(c) \\ c \sim \mu_c,\ (s_0,\dots) \sim \boldsymbol{\tau}(c)}} \sum_{t=0}^{H-1} \mathcal{I}(\boldsymbol{\pi}(s_t, c); \tilde{\boldsymbol{\pi}}(\tau, c) \mid c_{1:n}, \tau_{1:n}) \tag{62}$$

$$\leq (1 + \epsilon_n) \frac{2\bar{\sigma}^2}{\tau_{\min}^2} \max_{c' \in \mathcal{C}} \mathop{\mathbb{E}}_{\substack{\tau_{1:n-1} \sim \boldsymbol{\tau}(c_{1:n-1}) \\ \tau' \sim \boldsymbol{\tau}(c') \\ c \sim \mu_c,\ (s_0,\dots) \sim \boldsymbol{\tau}(c)}} \sum_{t=0}^{H-1} \mathcal{I}(\boldsymbol{\pi}(s_t, c); \tilde{\boldsymbol{\pi}}(\tau', c') \mid c_{1:n}, \tau_{1:n}) \tag{63}$$

$$= (1 + \epsilon_n) \frac{2\bar{\sigma}^2 \Gamma_n}{\tau_{\min}^2}. \tag{64}$$

where (i) follows from Lemma 8 and monotonicity of variance, (ii) holds as the state $s$ is within the support of $\boldsymbol{\tau}(\cdot|c)$, and (iii) follows from Lemma 1 as the difference between the expected mutual information and the mutual information for a realized trajectory is less than the difference in mutual information for two arbitrary realized trajectories. $\qquad\square$

This result can then be translated to the agnostic setting, for a regular policy $\pi^\star$, which we still model through the stochastic process $\boldsymbol{\pi}$. Without loss of generality we will assume that the prior variance is bounded by $\mathrm{Var}[\boldsymbol{\pi}(s, c)] \leq 1$.

**Lemma 4.** *(Well-calibrated confidence intervals, following Abbasi-Yadkori (2013)) Pick $\delta \in (0, 1)$. Assume that $\pi^\star$ lies in the RKHS $\mathcal{H}_k(\mathcal{C})$ of the kernel $k$ with norm $\|\pi^\star\|_k < \infty$, the noise $\epsilon_n$ is conditionally $\rho$-sub-Gaussian, and $\gamma_n$ is sublinear in $n$. Let $\beta_n(\delta) = \|\pi^\star\|_k + \rho\sqrt{2(\gamma_{(Hn)} + 1 + \log(1/\delta))}$. Then, for any $n > 1$ and $(s, c) \in \mathcal{S} \times \mathcal{C}$, $GP(\mu_n, k)$ is an all-time well-calibrated model of $\pi^\star$. Thus, jointly with probability at least $1 - \delta$,*

$$|\pi^\star(s, c) - \mu_n(s, c)| \leq \beta_n(\delta)\sigma_n.$$

We note that $\beta_n(\delta)$ depends on $\gamma_{(Hn)}$ as $Hn$ samples from the demonstrator's policy are collected up to round $n$. Combining Lemmas 3 and 4 we easily get for all $(s, c) \in \mathcal{S} \times \mathcal{C}$ and $n \geq 0$ with probability $1 - \delta$:

$$|\pi^\star(s, c) - \mu_n(s, c)| \stackrel{\text{Lemma 4}}{\leq} \beta_n(\delta)\sigma_n \stackrel{\text{Lemma 3}}{\leq} \beta_n(\delta)\left((1 + \epsilon_n)\frac{2\bar{\sigma}^2 \Gamma_n}{\tau_{\min}^2}\right)^{\frac{1}{2}} \tag{65}$$

While the analysis has so far dealt with a scalar $\pi^\star$, a simple union bound can guarantee that

$$\|\pi^\star(s, c) - \mu_n(s, c)\|_1 \leq \beta_n'(\delta)\|\bar{\sigma}\|_1\left((1 + \epsilon_n)\frac{2\Gamma_n}{\tau_{\min}^2}\right)^{\frac{1}{2}} \tag{66}$$

with probability at least $1 - \delta$ for an action space of dimension $|\mathcal{A}|$, where now $\beta_n'(\delta) = \|\pi^\star\|_k + \rho\sqrt{2(\gamma_{(Hn)} + 1 + \log(|\mathcal{A}|/\delta))}$. From now on, we will refer to $\mu_n$ as $\pi_n$. We are thus able to globally bound the $L_1$ distance of the imitator policy with respect to the expert policy with high probability under active fine-tuning.

It is clear that, even if this distance is small, the performance of an imitator which does not exactly match the expert ($\pi_n(s,c) \neq \pi^\star(s,c)$ for some $(s,c) \in \mathcal{S} \times \mathcal{C}$) can be arbitrarily low for arbitrary MDPs. It is however possible to show that, as long as the Q-function of the expert is smooth, the performance gap to the expert can be controlled. We note that, in case $\gamma L_P(1 + L_{\pi^\star}) < 1$, then the Q-function $Q^{\pi^\star}$ is guaranteed to be $L_Q$-Lipschitz continuous with $L_Q \leq \frac{L_R}{1 - \gamma L_P(1 + L_{\pi^\star})}$ (Rachelson & Lagoudakis, 2010). If smoothness holds, it is easy to connect the divergences in action space to performance gaps (Maran et al., 2023).

**Lemma 5.** *Let $\pi$ and $\pi'$ denote two deterministic policies. If the state-action value function $Q^{\pi'}$ is $L_{Q^{\pi'}}$-Lipschitz continuous, then:*

$$|J^\pi - J^{\pi'}| \leq \frac{L_{Q^{\pi'}}}{1 - \gamma} \mathbb{E}_{s \sim d^\pi}[\|\pi'(s,c) - \pi(s,c)\|_1].$$

*Proof.* Given a function $f : \mathcal{A} \to \mathbb{R}$, we denote the Lipschitz semi-norm $\|f(\cdot)\|_L = \sup_{a,a' \in \mathcal{A}} \frac{|(f(a)-f(a')|}{\|a-a'\|_2}$. We have:

$$J^\pi - J^{\pi'} \overset{(i)}{=} \frac{1}{1-\gamma} \mathbb{E}_{s \sim d^\pi}\left[ \mathbb{E}_{a \sim \pi(\cdot|s,c)}[A^{\pi'}(s,c,a)] \right] \tag{67}$$

$$= \frac{1}{1-\gamma} \mathbb{E}_{s \sim d^\pi}\left[ \mathbb{E}_{a \sim \pi(\cdot|s)}[Q^{\pi'}(s,c,a)] - V^{\pi'}(s,c) \right] \tag{68}$$

$$= \frac{1}{1-\gamma} \mathbb{E}_{s \sim d^\pi}\left[ \int_{a \in \mathcal{A}} \pi(a \mid s) Q^{\pi'}(s,c,a) - V^{\pi'}(s,c) \right] \tag{69}$$

$$= \frac{1}{1-\gamma} \mathbb{E}_{s \sim d^\pi}\left[ \int_{a \in \mathcal{A}} Q^{\pi'}(s,c,a)[\pi(a \mid s,c) - \pi'(a \mid s,c)] \right] \tag{70}$$

$$\leq \frac{1}{1-\gamma} \mathbb{E}_{s \sim d^\pi}\left[ \int_{a \in \mathcal{A}} \sup_{s,c \in \mathcal{S} \times \mathcal{C}} Q^\pi(s,c,a)[\pi(a \mid s,c) - \pi'(a \mid s,c)] \right] \tag{71}$$

$$\overset{(ii)}{\leq} \frac{1}{1-\gamma} \mathbb{E}_{s \sim d^\pi}\left[ \| \sup_{s,c \in \mathcal{S} \times \mathcal{C}} Q^{\pi'}(s,c,\cdot)\|_L \mathcal{W}(\pi(\cdot \mid s,c), \pi'(\cdot \mid s,c)) \right] \tag{72}$$

$$\overset{(iii)}{\leq} \frac{L_{Q^{\pi'}}}{1-\gamma} \mathbb{E}_{s \sim d^\pi}[\mathcal{W}(\pi(\cdot \mid s,c), \pi'(\cdot \mid s,c))] \tag{73}$$

$$\overset{(iv)}{=} \frac{L_{Q^{\pi'}}}{1-\gamma} \mathbb{E}_{s,c \sim d^\pi}[\|\pi(\cdot \mid s,c) - \pi'(\cdot \mid s,c)\|_1] \tag{74}$$

where (i) follows from the performance difference lemma (Kakade & Langford, 2002), (ii) follows from the definition of $L_1$ Wasserstein distance, (iii) holds as $L_{Q^\pi} \geq \|\sup_{s,c \in \mathcal{S} \times \mathcal{C}} Q^\pi(s,c,\cdot)\|_L$ and (iv) follows from both policies being deterministic. The proof is completed by taking the absolute value on both sides. $\square$

So far, we have shown rates of convergence for the imitator, and connected its error to performance. Our main formal result can be shown by coordinating the lemmas so far presented.

**Theorem 3.** *(Performance guarantees for active multi-task BC) Let Assumptions 2, 3 and 4 hold. Pick $\delta \in (0,1)$. Assume that $\pi^\star$ lies in the RKHS $\mathcal{H}_k(\mathcal{C})$ of the kernel $k$ with norm $\|\pi^\star\|_k < \infty$, the noise $\epsilon_n$ is conditionally $\rho$-sub-Gaussian, and $\gamma_n$ is sublinear in $n$. If each demonstrated task is selected according to the criterion in Equation 1, then with probability at least $1 - \delta$ the performance difference between the expert policy $\pi^\star$ and the imitator policy $\pi_n$ after $n$ demonstrations can be upper bounded:*

$$J^{\pi^\star} - J^{\pi_n} \leq \frac{\sqrt{2} L_{Q^{\pi^\star}} \|\bar{\sigma}\|_1}{\tau_{min}(1-\gamma)} \left( (1 + \epsilon_n)\beta_n^{'2}(\delta)\Gamma_n \right)^{\frac{1}{2}} = O(\gamma_{(Hn)})/\sqrt{n},$$

*where* $\epsilon_n = 8H^{\frac{3}{2}}(1 + \max(L_\pi, L_P))^H \max(\epsilon_0, \epsilon_\pi, \epsilon_P)$. *Furthermore, if* $\gamma_n = O(\log n)$ *(e.g., for linear kernels), then* $J^{\pi^\star} - J^\pi \overset{n\to\infty}{\to} 0$.

*Proof.*

$$J^{\pi^\star} - J^\pi \overset{(i)}{=} |J^\pi - J^{\pi^\star}| \tag{75}$$

$$\overset{\text{Lemma 5}}{\leq} \frac{L_{Q^{\pi^\star}}}{1-\gamma} \mathbb{E}_{s\sim d^{\pi^\star}} [\|\pi_n(s,c) - \pi^\star(s,c)\|_1] \tag{76}$$

$$\overset{\text{Lemma 3,4}}{\leq} \frac{L_{Q^{\pi^\star}}}{1-\gamma} \cdot \beta'_n(\delta)\|\bar\sigma\|_1 \Big((1+\epsilon_n)\frac{2\Gamma_n}{\tau_{\min}^2}\Big)^{\frac{1}{2}} \tag{77}$$

$$= \frac{\sqrt{2}L_{Q^{\pi^\star}}\|\bar\sigma\|_1}{\tau_{\min}(1-\gamma)} \Big((1+\epsilon_n)\beta_n'^2(\delta)\Gamma_n\Big)^{\frac{1}{2}} \tag{78}$$

where (i) is due to the fact that $J^{\pi^\star} \geq J^\pi$ for any policy $\pi$, and the expectation fades due to uniform convergence. The only terms with a dependency on $n$ are $\beta'_n(\delta) = O(\gamma_{(Hn)}^{\frac{1}{2}})$ and $\Gamma_n = O(\gamma_{(Hn)})/n$, which can be combined in the asymptotic notation in the Theorem. If $\gamma_n = O(\log n)$, then $J^{\pi^\star} - J^\pi = O(\log n)/\sqrt{n} \overset{n\to\infty}{\to} 0$. For a summary of magnitudes of $\gamma_n$ for common kernels, we refer to Table 3 in Hübotter et al. (2024b). $\qquad\square$

# B  GUARANTEES IN NON-LIPSCHITZ MDPS

The main result reported in Theorem 3 provides anytime guarantees on the agent's performance, assuming smoothness in the MDP. However, it is possible to replace this assumption with a weaker one, at the cost of only retaining guarantees in expectation. This weaker version of the theorem can be retrieved by simply assuming smoothness on the *noise*, rather than on the MDP, and leveraging results recently presented by Maran et al. (2023).

**Assumption 5.** *The noise distribution $\epsilon$ is $L_\ell$-TV-Lipschitz continuous.*

This assumption is satisfied by a large class of Gaussian and sub-Gaussian distributions (Maran et al., 2023). We can build upon Assumption 4 and Lemma 2, and start by providing a weaker version of Lemma 3.

**Lemma 6.** *(Uniform convergence of marginal variance in expectation) Under Assumption 1, for any $n \geq 0$ and $(s,c) \in \mathcal{S} \times \mathcal{C}$,*

$$\mathbb{E}_{\tau_{1:n-1}\sim\boldsymbol{\tau}(c_{1:n-1})} \sigma_n^2(s,c) \leq \frac{2\bar\sigma^2\Gamma_n}{\tau_{min}^2},$$

*where $\tilde\sigma^2 = \max_{(s,c)\in\mathcal{S}\times\mathcal{C}} \sigma_0^2(s,c) + \rho^2(s,c)$ and $\tau_{min} = \min_{s,c\in\mathcal{S}\times\mathcal{C}} \mathbb{E}_{\tau\sim\boldsymbol{\tau}(c)} \mathbf{1}_{s\in\tau}$.*

*Proof.* We resume from Inequality 60 in the proof of Lemma 3:

$$\sigma_n^2(s,c) \leq \frac{2\bar\sigma^2}{\tau_{\min}^2} \mathop{\mathbb{E}}_{\substack{c\sim\mu_c \\ \tau\sim\boldsymbol{\tau}(c) \\ (s_0,\dots)\sim\boldsymbol{\tau}(c)}} \sum_{t=0}^{H-1} \mathcal{I}(\boldsymbol{\pi}(s_t,c); \tilde{\boldsymbol{\pi}}(\tau,c) \mid c_{1:n},\tau_{1:n}) \tag{79}$$

Therefore,

$$\mathbb{E}_{\tau_{1:n-1} \sim \boldsymbol{\tau}(c_{1:n-1})} \sigma_n^2(s,c) \leq \mathbb{E}_{\substack{\tau_{1:n-1} \sim \boldsymbol{\tau}(c_{1:n-1}) \\ \tau \sim \boldsymbol{\tau}(c) \\ c \sim \mu_c, (s_0,\dots) \sim \boldsymbol{\tau}(c)}} \frac{2\bar{\sigma}^2}{\tau_{\min}^2} \sum_{t=0}^{H-1} \mathcal{I}(\boldsymbol{\pi}(s_t,c); \tilde{\boldsymbol{\pi}}(\tau,c) \mid c_{1:n}, \tau_{1:n}) \tag{80}$$

$$\leq \max_{c' \in \mathcal{C}} \mathbb{E}_{\substack{\tau_{1:n-1} \sim \boldsymbol{\tau}(c_{1:n-1}) \\ \tau' \sim \boldsymbol{\tau}(c') \\ c \sim \mu_c, (s_0,\dots) \sim \boldsymbol{\tau}(c)}} \frac{2\bar{\sigma}^2}{\tau_{\min}^2} \sum_{t=0}^{H-1} \mathcal{I}(\boldsymbol{\pi}(s_t,c); \tilde{\boldsymbol{\pi}}(\tau',c') \mid c_{1:n}, \tau_{1:n}) \tag{81}$$

$$= \frac{2\bar{\sigma}^2 \Gamma_n}{\tau_{\min}^2}. \tag{82}$$

$\square$

Having bounded variance at each round, this time in expectation, we can invoke Lemma 4 to bound the expected distance to the optimal policy with high probability:

$$\mathbb{E}_{\tau_{1:n-1} \sim \boldsymbol{\tau}(c_{1:n-1})} \|\pi^\star(s,c) - \mu_n(s,c)\|_1 \leq \beta_n'(\delta) \|\bar{\sigma}\|_1 \left(\frac{2\Gamma_n}{\tau_{\min}^2}\right)^{\frac{1}{2}} \tag{83}$$

Instead of leveraging bounds for the imitator's performance in Lipschitz-smooth settings, we can instead use the fact that the expert's actions are corrupted by smooth noise. In this setting, it is instead possible to control the suboptimality of the imitator with respect to the *noisy* expert. We report the following Theorem from Maran et al. (2023), and refer to the original work for the proof.

**Lemma 7.** *Let $\pi^\star$, $\tilde{\boldsymbol{\pi}}$ and $\pi$ denote the expert, noisy expert and imitator policy, respectively. If Assumption 5 holds, then:*

$$J^{\tilde{\boldsymbol{\pi}}} - J^\pi \leq \frac{2L_\ell Q_{max}}{1-\gamma} \mathbb{E}_{s \sim \mu_{\tilde{\boldsymbol{\pi}}}}[\mathcal{W}(\pi^\star(\cdot \mid s), \pi(\cdot \mid s))],$$

*where $Q_{max} = \max_{(s,c,a) \in \mathcal{S} \times \mathcal{C} \times \mathcal{A}} |Q(s,a)|^\pi$ and $\mathcal{W}$ represents the Wasserstein 1-distance.*

As the expert $\pi^\star$ and the imitator $\pi_n$ are both deterministic, this implies that

$$J^{\tilde{\boldsymbol{\pi}}} - J_n^\pi \leq \frac{2L_\ell Q_{\max}}{1-\gamma} \mathbb{E}_{s \sim \mu_{\tilde{\boldsymbol{\pi}}}} \|(\pi^\star(\cdot \mid s), \pi(\cdot \mid s))\|_1. \tag{84}$$

By invoking this Lemma, we can thus conclude that, with probability at least $1 - \delta$

$$\mathbb{E}_{\tau_{1:n-1} \sim \boldsymbol{\tau}(c_{1:n-1})} J^{\tilde{\boldsymbol{\pi}}} - J^{\pi_n} \leq \frac{2^{\frac{3}{2}} L_\ell Q_{\max} \|\bar{\sigma}\|_1}{\tau_{\min}(1-\gamma)} \beta_n'(\delta) \Gamma_n^{\frac{1}{2}} = O(\gamma_{(Hn)} n^{-\frac{1}{2}}). \tag{85}$$

Therefore, if $\gamma_n = O(\log n)$, then $\mathbb{E}_{\tau_{1:n-1} \sim \boldsymbol{\tau}(c_{1:n-1})} J^{\tilde{\boldsymbol{\pi}}} - J^\pi = O(\frac{\log n}{\sqrt{n}}) \overset{n \to \infty}{\to} 0$. While these performance guarantees only hold in expectation, they arise from minimal assumptions, mostly regarding the policy class and the perturbation noise, and can thus be applied to arbitrary MDPs.

## C    PRACTICAL OBJECTIVE

Following up on the approximations reported in Section 4.2, we present the empirical estimate of the objective that is used through experiments. In particular, we show how the expectations in Equation 2 may be approximated with finite samples. The original criterion is expressed as

$$c_n = \arg\min_{c' \in \mathcal{C}} \phi_n(c') = \arg\min_{c' \in \mathcal{C}} \mathbb{E}_{\substack{\tau_{1:n-1} \sim \boldsymbol{\tau}(c_{1:n-1}), \tau' \sim \boldsymbol{\tau}(c') \\ c \sim \mu_c, (s_0,\dots) \sim \boldsymbol{\tau}(c)}} \sum_{t=0}^{H-1} \mathcal{H}(\boldsymbol{\pi}(s_t,c) \mid c', \tau', c_{1:n-1}, \tau_{1:n-1}).$$

$$\tag{86}$$

An empirical estimate can be derived as follows:

$$\phi_n(c') = \mathop{\mathbb{E}}_{\substack{\tau_{1:n-1}\sim\boldsymbol{\tau}(c_{1:n-1}),\ \tau'\sim\boldsymbol{\tau}(c')\\ c\sim\mu_c,\ (s_0,\dots)\sim\boldsymbol{\tau}(c)}} \sum_{t=0}^{H-1} \mathcal{H}(\boldsymbol{\pi}(s_t,c)\mid c',\tau',c_{1:n-1},\tau_{1:n-1}) \tag{87}$$

$$\stackrel{(i)}{\approx} \mathop{\mathbb{E}}_{\substack{\tau'\sim\boldsymbol{\tau}(c')\\ c\sim\mu_c,\ (s_0,\dots)\sim\boldsymbol{\tau}(c)}} \sum_{t=0}^{H-1} \mathcal{H}(\boldsymbol{\pi}(s_t,c)\mid c',\tau',c_{1:n-1},\hat{\tau}_{1:n-1}) \tag{88}$$

$$\stackrel{(ii)}{\approx} \frac{1}{|\hat{\mathcal{C}}|}\sum_{c\in\hat{\mathcal{C}}} \mathop{\mathbb{E}}_{\substack{\tau'\sim\boldsymbol{\tau}(c')\\ (s_0,\dots)\sim\boldsymbol{\tau}(c)}} \sum_{t=0}^{H-1} \mathcal{H}(\boldsymbol{\pi}(s_t,c)\mid c',\tau',c_{1:n-1},\tau_{1:n-1}) \tag{89}$$

$$= \frac{1}{|\hat{\mathcal{C}}|}\sum_{c\in\hat{\mathcal{C}}} \mathop{\mathbb{E}}_{\substack{\tau'\sim\hat{\boldsymbol{\tau}}\\ (s_0,\dots)\sim\hat{\boldsymbol{\tau}}}} \frac{\boldsymbol{\tau}(\tau'|c')}{\hat{\boldsymbol{\tau}}(\tau')}\frac{\boldsymbol{\tau}((s_0,\dots)|c)}{\hat{\boldsymbol{\tau}}((s_0,\dots))} \sum_{t=0}^{H-1} \mathcal{H}(\boldsymbol{\pi}(s_t,c)\mid c',\tau',c_{1:n-1},\tau_{1:n-1}) \tag{90}$$

$$= \frac{1}{|\hat{\mathcal{C}}|}\sum_{c\in\hat{\mathcal{C}}} \mathop{\mathbb{E}}_{\substack{\tau'\sim\hat{\boldsymbol{\tau}}\\ (s_0,\dots)\sim\hat{\boldsymbol{\tau}}}} w(\tau',c')w((s_0,\dots),c) \sum_{t=0}^{H-1} \mathcal{H}(\boldsymbol{\pi}(s_t,c)\mid c',\tau',c_{1:n-1},\tau_{1:n-1}) \tag{91}$$

$$\stackrel{(iii)}{\approx} \frac{1}{|\hat{\mathcal{C}}|(n-1)^2}\sum_{c\in\hat{\mathcal{C}}}\sum_{\substack{\tau'\in\hat{\tau}_{1:n-1}\\ (s_0,\dots)\in\hat{\tau}_{1:n-1}}} w(\tau',c')w((s_0,\dots),c) \sum_{t=0}^{H-1} \mathcal{H}(\boldsymbol{\pi}(s_t,c)\mid c',\tau',c_{1:n-1},\tau_{1:n-1}),$$
$$\tag{92}$$

where (i) uses a single sample to estimate the expectation over past trajectories, (ii) uses a sample-based approximation to the target task distribution $\mu_c$, and (iii) uses the importance sampling trick introduced in Section 4.2, with $w(\tau,c) = \frac{(n-1)\prod_{t=0}^{H-1}\tilde{\boldsymbol{\pi}}(a_t|s_t,c)}{\sum_{i=0}^{n-1}\prod_{t=0}^{H-1}\tilde{\boldsymbol{\pi}}(a_t|s_t,c_i)}$. This final approximate objective does not involve expectations, and can be efficiently computed. The complexity of evaluating the criterion for a single task $c'$ scales linearly with the number of samples in $\hat{\mathcal{C}}$ and quadratically with the number of rounds $n$. However, the dependency on the number of rounds can be removed by evaluating the second sum over a fixed number of trajectories sampled among $\tau_{1:n-1}$, ensuring that the complexity does not depend on the round.

## D    ADDITIONAL RESULTS FOR AMF-GP

Figure 3 only reports full return curves for two representative pre-training settings, namely those involving 6/12 and 12/12 demonstrated tasks. We here report full results for each task allocation, spanning from 1/12 to 12/12 demonstrated tasks. For each setting, we report both average multi-task return and average policy entropy curves.

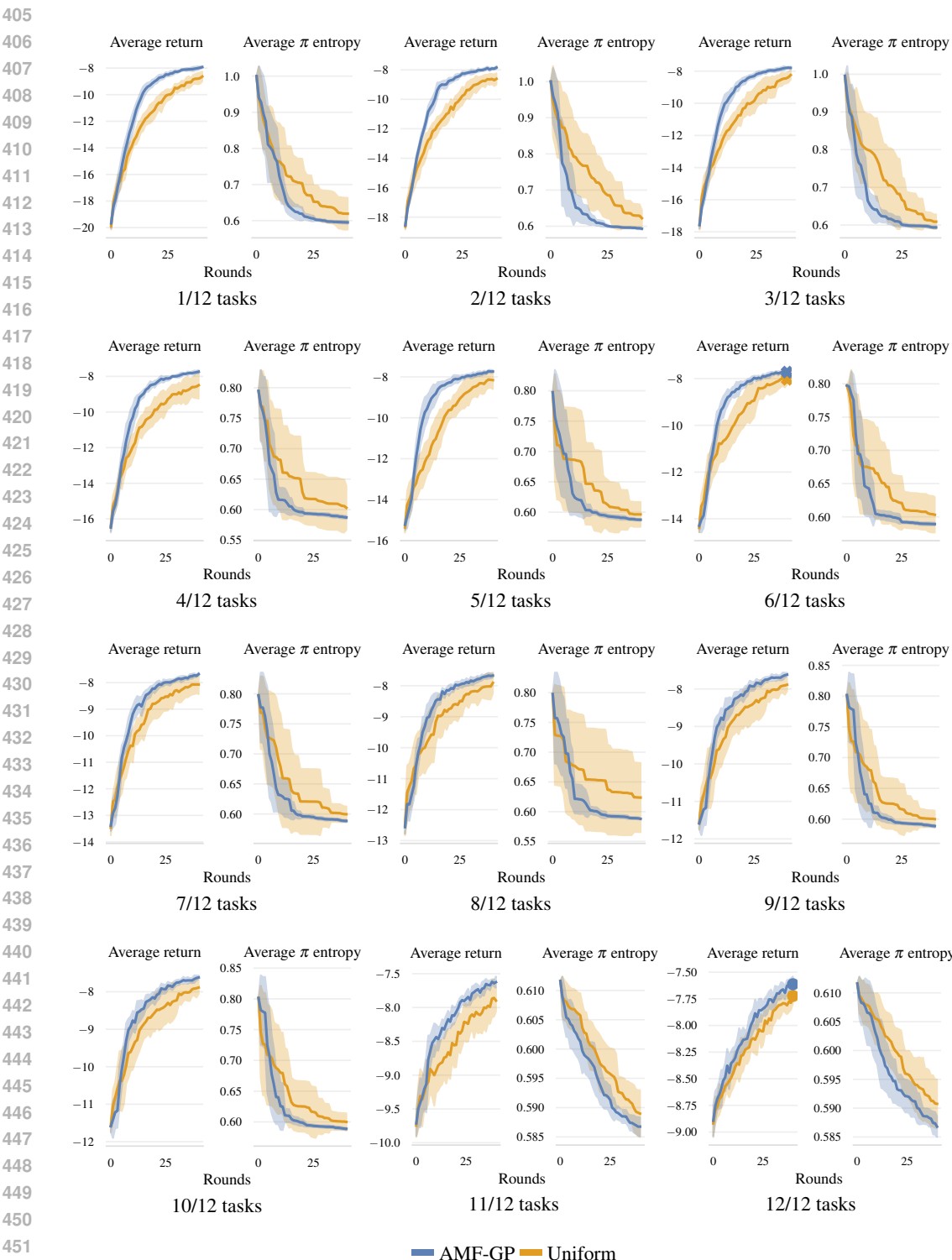

Figure 8: Additional results in GP settings for a 2D integrator (see Figure 2). AMF-GP results in improved sample efficiency across all pre-training regimes, and is particularly effective for skewed pre-training distributions (e.g., when pre-training demonstrations have been allocated to 1/12 or 6/12 tasks).

# E ADDITIONAL RESULTS FOR AMF-NN

Results in Figure 4 are computed over two representative pre-training distributions: one allocating pre-training demonstrations uniformly over all tasks, the other one only demonstrating the first two tasks. We report these results again, and compare them with those for several other pre-training distributions. In particular, we evaluate a family of skewed priors which are only trained on one or two tasks. Results are reported for FrankaKitchen in Figure 9 and for Metaworld in Figure 10, and are consistent with patterns observed in Figure 4.

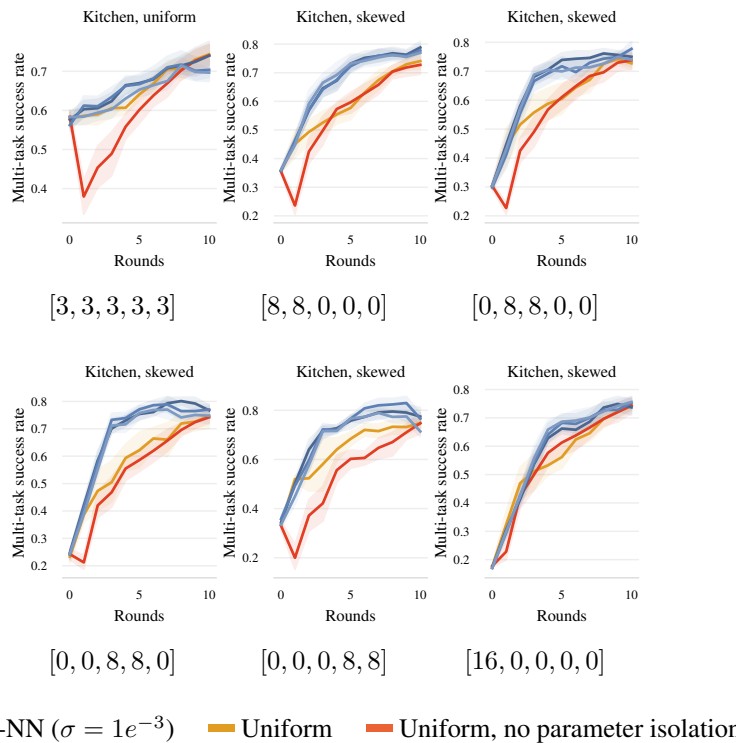

Figure 9: Additional results for AMF-NN in FrankaKitchen with state inputs. We evaluate several allocations of the pre-training demonstrations, as labeled below each plot (e.g., the label $[8, 8, 0, 0, 0]$ indicates that 8 demonstrations were provided for each of the first two tasks each, and none for the remaining tasks).

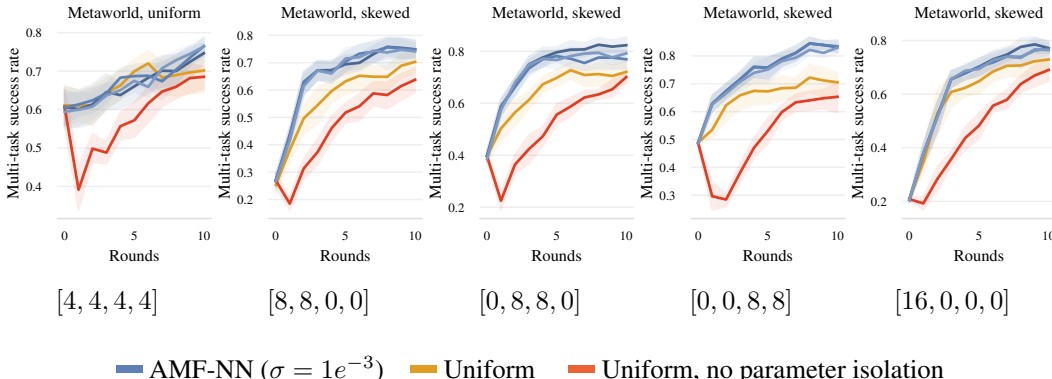

Figure 10: Additional results for AMF-NN in Metaworld with state inputs. We evaluate several allocations of the pre-training demonstrations, as labeled below each plot (e.g., the label $[8, 8, 0, 0]$ indicates that 8 demonstrations were provided for each of the first two tasks each, and none for the remaining tasks).

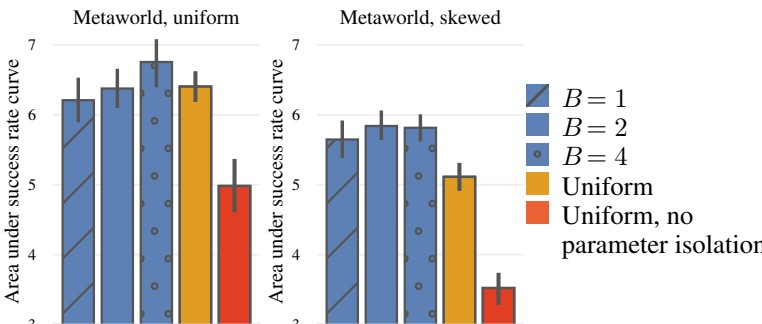

Figure 11: Influence of batch size $B$ over area under success rate curve with a budget of 20 demonstrations in Metaworld.

Moreover, we also report an ablation over the choice of batch size for Metaworld in Figure 11, thus complementing the one reported in Section 5.3. We observe a slight upward trend favoring larger batch size in the case of an uniform prior, but smaller batches remain overall desirable.

## F UNCERTAINTY ABLATION

Section 5.4 evaluates alternative uncertainty quantification schemes in FrankaKitchen for two pre-training distributions. This Section extends these results to include results for Metaworld, and for several other pre-training settings (see Figures 12 and 13). Results are consistent with those so far reported, suggesting that loss gradient embeddings are an important component for the empirical performance of AMF-NN.

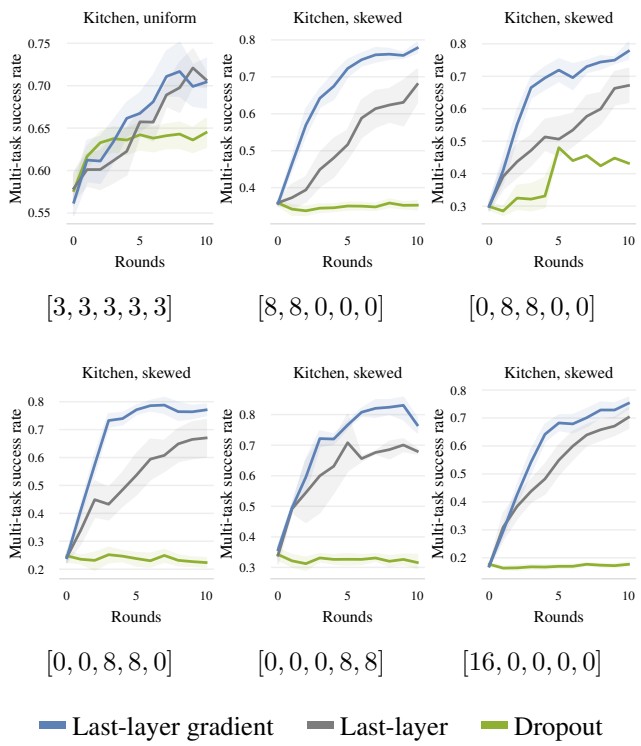

Figure 12: Additional results for AMF-NN in FrankaKitchen with state inputs and different uncertainty quantification techniques. Task allocation during pre-training is reported under each plot.

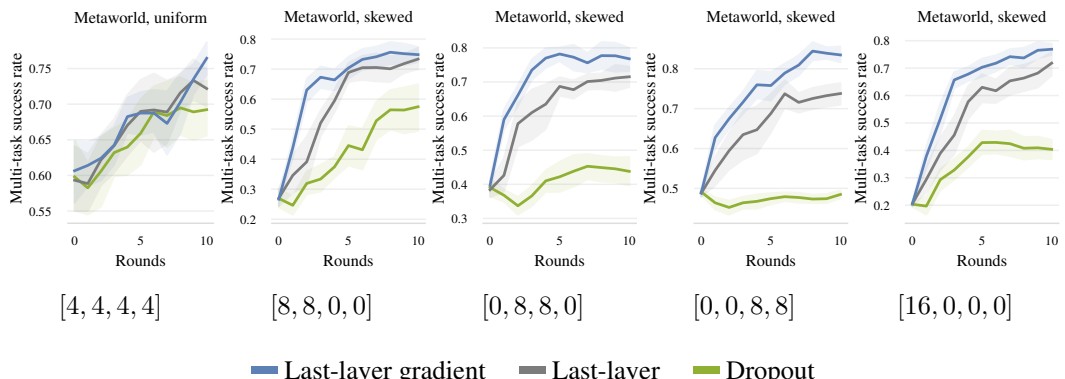

Figure 13: Additional results for AMF-NN in Metaworld with state inputs and different uncertainty quantification techniques. Task allocation during pre-training is reported under each plot.

## G  MITIGATING FORGETTING

The ability of neural networks to adapt to shifts in training distribution while retaining information is an important object of interest in lifelong and continual learning (Wang et al., 2024b). In general, learned models display a trade-off between their ability to integrate novel information, and their memory of previously observed training samples. Arguably, common neural network architectures can easily fit new data (save for loss of plasticity (Lyle et al., 2023)), but are known to forget previous information, often catastrophically. This problem is of utmost relevance in our setting, in which the pre-trained network is not just leveraged as a useful initialization, but may already capable of solving some tasks. Hence, the fine-tuning procedure should be careful not to disrupt this ability.

Several methods aimed at mitigating forgetting can be traced back to rehearsal (Riemer et al., 2019; Chaudhry et al., 2019) and regularization (Kirkpatrick et al., 2017) strategies. While rehearsal approaches are often effective, they also require access to pre-training data, which is unrealistic in our setting. Hence we consider two common regularization technique, namely L2-regularization to the pre-trained weights, and EWC (Kirkpatrick et al., 2017). The latter can be seen as a more nuanced version of the former, which adaptively scales the regularization strength according to the curvature of the loss landscape.

Furthermore, we consider a continual learning algorithm based on Git Re-Basin (Ainsworth et al., 2023), which was originally proposed as a model-merging technique that seeks linearly mode connected (LMC) areas in the loss landscape by permuting network weights. Interestingly, while much of the following work additionally relies on rehearsal techniques (Peña et al., 2023; Wang et al., 2024a), Ainsworth et al. (2023) also propose a data-independent matching algorithm, which can be applied to our setting. In practice, after each round, we apply the permutation returned by Git Re-Basin to the updated policy's weights, as if we had to merge it with the policy weights at the

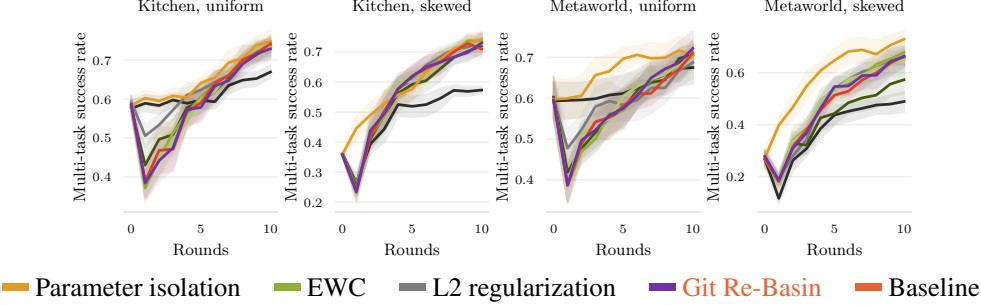

Figure 14: Performance of uniform task sampling with several techniques to mitigate forgetting. Darker shades represent stronger regularization coefficients. L2, EWC regularization and Git Re-Basin are not effective in this setting.

previous round. This should return updated weights lying in a LMC area with respect to the policy's previous weights, mitigating the performance gap with respect to the pre-training objective.

Unfortunately, we find these methods to be insufficient in our setting, as reported in Figure 14. When coupled with large regularization weights, the asymptotical performance of L2-regularization and EWC is significantly limited. When regularization weights are too low, they recover the performance of a naive baseline. Intermediate values were found to interpolate between the two behaviors, without addressing the forgetting issue. On the other hand, we fine that fine-tuning updates did not cause large shifts in the policy weights, therefore permutations explored by Git Re-Basin would hardly induce changes in the parameters. While less scalable, we found hard parameter isolation to be the only effective solution amongst the one we tested. As mitigating forgetting in neural network is an orthogonal direction to the main topic of this work, we adopt this solution, and expect that further developments in continual learning will be applicable in our setting.

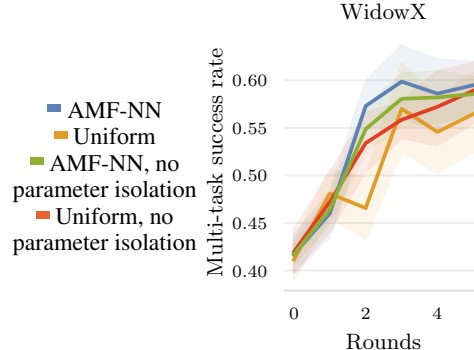

Finally, we remark that preventing forgetting is crucial for fine-tuning, irrespectively of the data collection strategy used. For completeness, we present extended results from Figure 4 in Figure 16. In particular, we include learning curves for AMF without parameter isolation, showing that the performance of all data collection strategies drops to comparable levels if the continual learning problem is not addressed. We also present extended results from Figure 7 in Figure 15. In this case, the policy is parameterized by a much larger models. We confirm that, as the model scale increases, catastrophic forgetting is partially alleviated, independently from the data collection strategy. This is consistent with trends in language modeling (Ramasesh et al., 2022). While, in this case, parameter isolation is not entirely necessary, catastrophic forgetting remains a pressing problem for datasets and models of modest size.

Figure 15: Evaluation on WidowX tasks without parameter isolation.

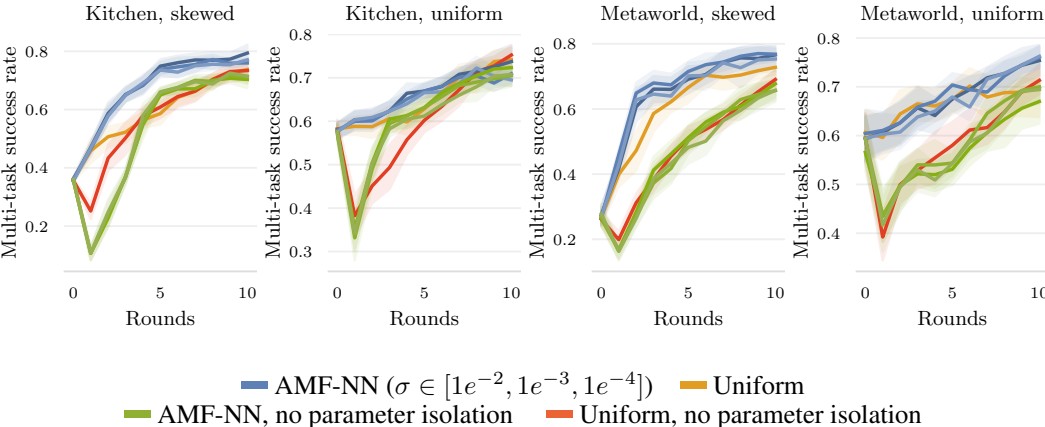

Figure 16: Extended results from Figure 4, including performance of AMF-NN without parameter isolation. If catastrophic forgetting is not addressed, AMF recovers the performance of a uniform data collection strategy.

## H    DOES AMF REBALANCE DEMONSTRATION COUNTS?

In discrete task spaces, counting the number of demonstrations for each task is possible. In this case, a naive data selection strategy would simply request demonstrations for tasks that have been demonstrated the least in the past. If all tasks require a similar amount of demonstrations, this would empirically perform very well. In our setting, however, data selection algorithms do not have knowledge of pre-training data. For this reason, a count could only be kept with respect to the fine-

tuning demonstrations: actively balancing this count would lead to a near-uniform task selection, and recover the performance of uniform sampling in expectation.

Nevertheless, we implement this "rebalancing" criterion as a *privileged* baseline, which assumes access to the pre-training task distribution. We evaluate it in the standard settings for AMF-NN from Figure 4. In Figure 17, we observe that AMF-NN is able to match the performance of this baseline in skewed settings, or outperform it in uniform settings, **despite having no knowledge of the pretraining distribution.**

This implies that AMF can infer information on the pre-training phase through estimation of the policy's uncertainty, and is capable of automatically recovering a "rebalancing" strategy. Moreover, AMF-NN considers the reduction in entropy across several tasks: hence, it can outperform the "rebalancing" baseline by focusing on tasks that are harder to learn or that could, in principle, lead to learning progress on other tasks. Further empirical evidence for these behaviors is shown in Appendix I.

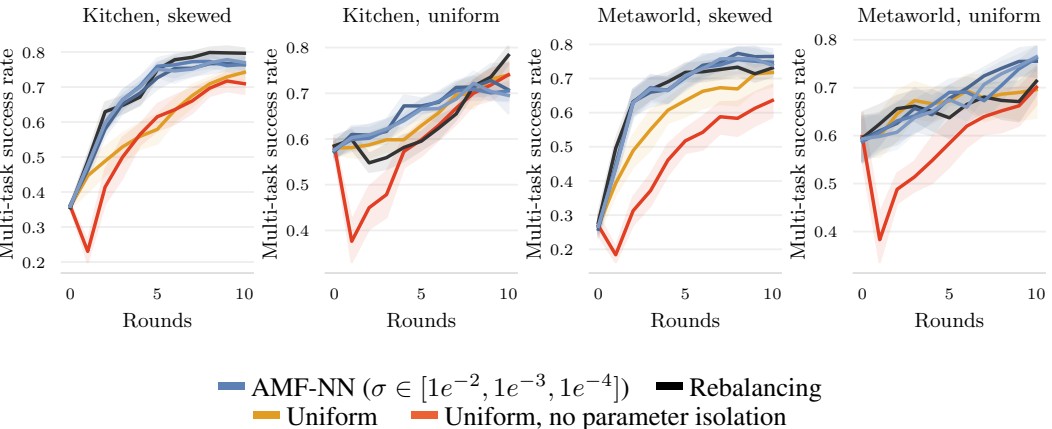

Figure 17: Extended results from Figure 4, including a privileged "rebalancing" baseline.

## I SINGLE-TASK PERFORMANCE

This Section presents a detailed look at the data selection strategies induced by AMF-NN. For this purpose, we consider the main experiments in Kitchen and Metaworld outlined in Figure 4, and plot single-task success rates, as well as the amount of demonstrations collected over time.

In the case of skewed pre-training (Fig. 18 and 20), we observe that AMF samples tasks that were not present in the pretraining dataset more often, **without having access to any direct information on the pre-training distribution**. Moreover, even if multiple tasks have the same frequency in the pre-training distribution, AMF will prefer the ones that induce a larger reduction in posterior uncertainty: for instance, in Kitchen, AMF selects the harder task `Left door` more often. Similarly, in the uniform pre-training case (see Fig. 19 and 21), AMF does not simply sample tasks uniformly. Rather, it focuses on those that maximize learning process (e.g., `Left door`, `Microwave`), while largely ignoring tasks that are nearly learned (e.g., `Knob off`, `Sliding door`). As a consequence, it can outperform naive baselines (see Appendix H).

We remark that these task selection strategies arise naturally from our information-based criterion in Equation 1, without any direct information on the pre-training distribution, nor any explicit policy evaluation.

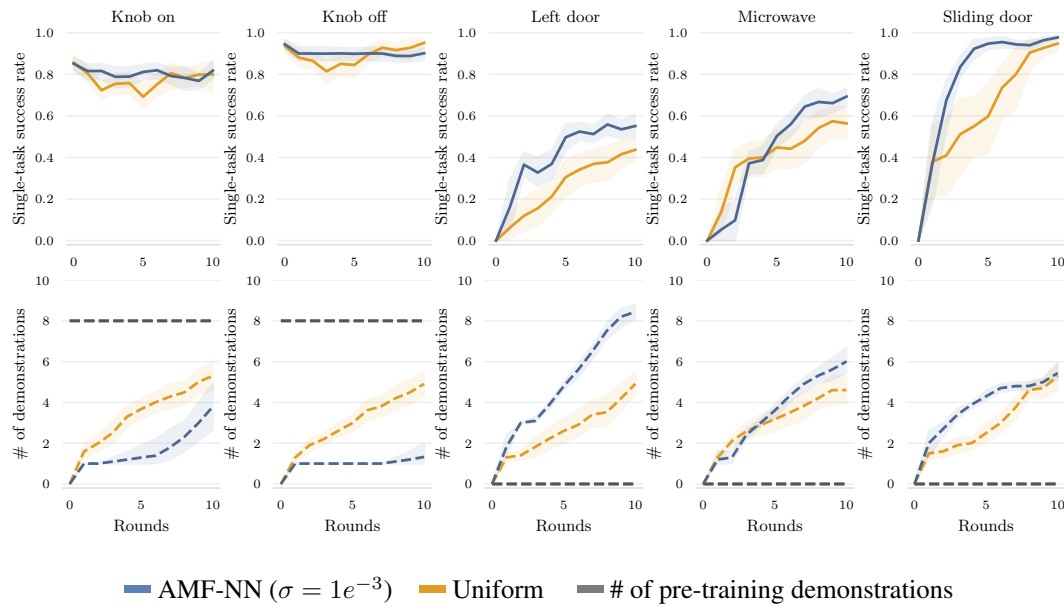

Figure 18: Single-task curves for skewed pre-training in Kitchen. Dashed lines represent demonstrations counts, with grey lines displaying the (inaccessible) count of pre-training demonstrations.

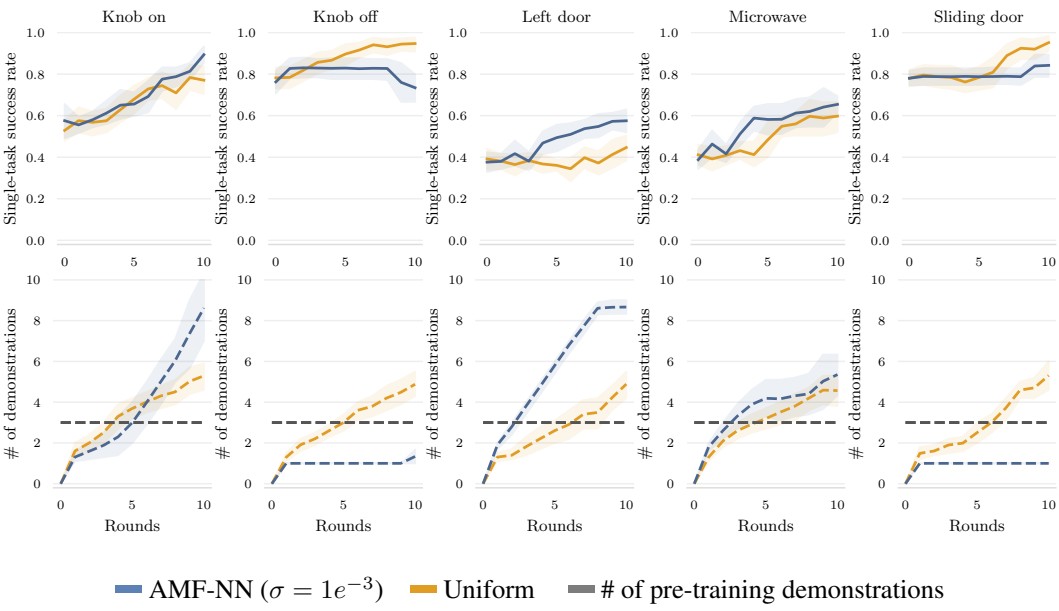

Figure 19: Single-task curves for uniform pre-training in Kitchen. Dashed lines represent demonstrations counts, with grey lines displaying the (inaccessible) count of pre-training demonstrations.

## J  ANALYSIS OF IMPORTANCE WEIGHTS

Importance weights (as introduced in Equation 3) allow estimating the expert's occupancy for arbitrary tasks. Naturally, the quality of importance weights depends on many factors, including the dimensionality of the trajectory space, and the density with which available data covers it. In this section, we report a qualitative evaluation of importance weights for both AMF-GP and AMF-NN (Figures 22 and 23, respectively). In both cases, we find that informative weights can be retrieved eventually, given the proper amount of clipping (as described in Appendix L). While in early rounds

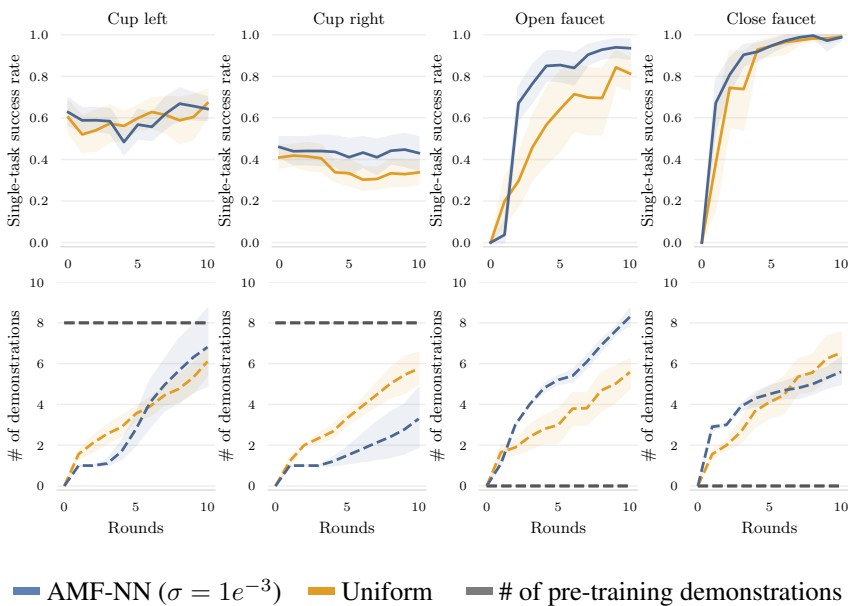

Figure 20: Single-task curves for skewed pre-training in Metaworld. Dashed lines represent demonstrations counts, with grey lines displaying the (inaccessible) count of pre-training demonstrations.

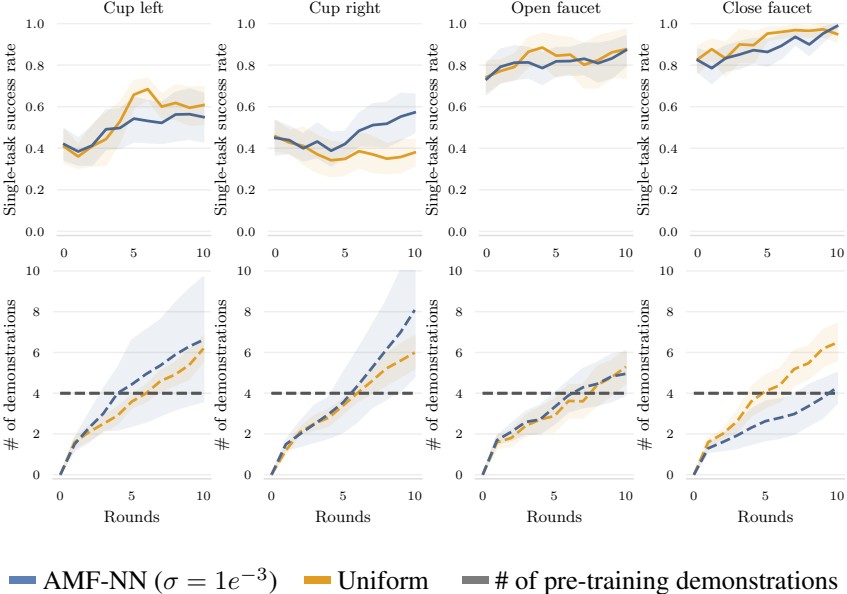

Figure 21: Single-task curves for uniform pre-training in Metaworld. Dashed lines represent demonstrations counts, with grey lines displaying the (inaccessible) count of pre-training demonstrations.

of the algorithm, weights can be inaccurate, leading to a poor estimate of the objective, we observe that the quality of importance sampling weights improves within a handful of rounds.

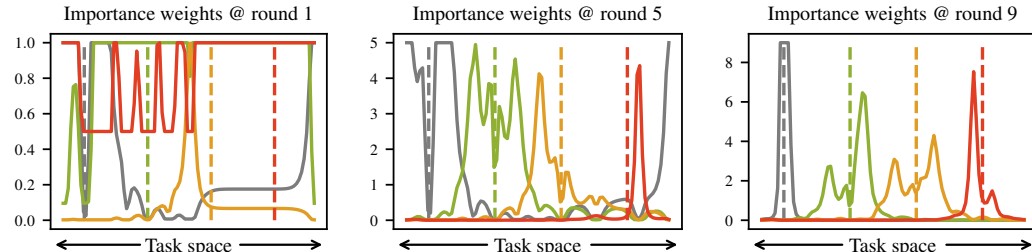

Figure 22: Analysis of importance sampling weights for AMF-GP. We consider the skewed pretraining setting from Figure 3, and compute importance weights after 1, 5 and 9 rounds. We sample four tasks $c_{0:3}$, represented by vertical dashed lines of different colors. For each task $c_i$, we collect a demonstration $\tau_i$ and sweep over $c' \in \mathcal{C}$ on the x-axis; we plot $w(\tau_i, c')$ with solid lines. We observe that importance weights are uninformative in early parts of training, but converge to more accurate values within a few rounds.

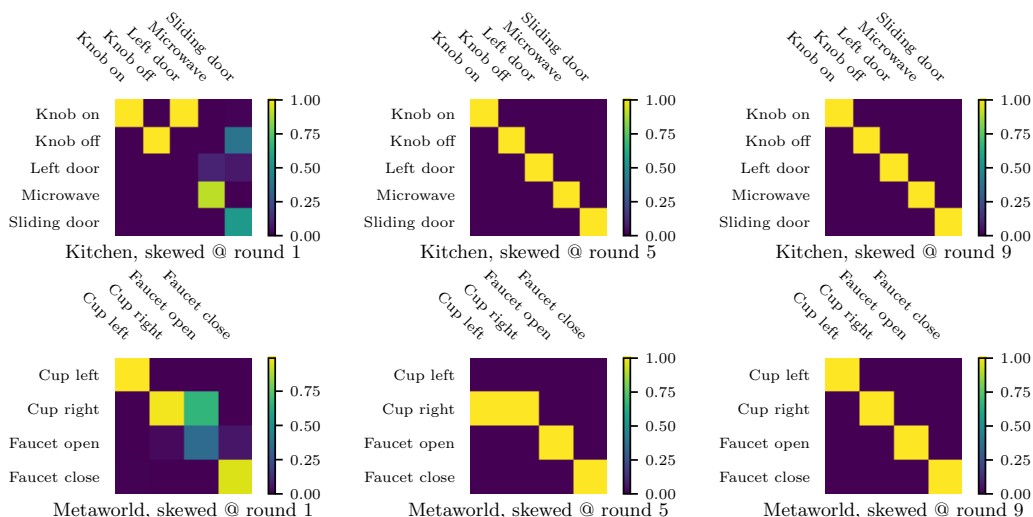

Figure 23: Analysis of importance sampling weights for AMF-NN. We consider the skewed pretraining setting from Figure 4, and compute importance weights after 1, 5 and 9 rounds. We visualize weights for both Kitchen (top) and Metaworld (bottom). As the task set is discrete, we consider all tasks ($c_i \in \mathcal{C}$), and collect one demonstration $\tau_i$ for each. The entry of each colormap at row $i$ and column $j$ represents $w(\tau_j, c_i)$. Again, we observe that at the beginning of training importance weights can be inaccurate, particularly for tasks $c_i$ that have not been sufficiently demonstrated. However, as more data is collected and the policy specializes to each task, the weights converge.

## K   CRITERION VS RETURNS

As a didactic example, we evaluate the criterion optimized by AMF-GP in a particular instance. We adopt the settings presented in Section 5.1, and pre-train a GP policy by providing 50 demonstrations in the 2D integrator environment, uniformly sampled among tasks in the top half of the target circle. We represent the task space along one dimension, and plot the smoothed pre-training distribution on the top of Figure 24. The second row of the Figure displays the evaluation of the criterion in Equation 2 for 100 tasks uniformly sampled across the entire task space. By comparison with the plot above, it is evident that the criterion is significantly lower for tasks that have not yet been demonstrated. These tasks are also those that, if demonstrated, would lead to a greater increase in multi-task performance after fine-tuning, as reported in the bottom row of Figure 24. In this instance, it's easy to see that the criterion leads to selection of tasks which have not been demonstrated sufficiently, and that will thus lead to greater policy performance.

Figure 24: Didactic example on correlation between pre-training distribution over tasks (top), evaluations of the AMF criterion for each task (middle) and return after fine-tuning on a demonstration for a given task (bottom).

## L   IMPLEMENTATION DETAILS

In order to ease reproducibility, we open-source our codebase on the project's anonymous website.[4] Furthermore, we describe several implementation details in the following sections.

### L.1   METRICS

All metrics are reported in the form of their mean and the 90% simple bootstrap confidence intervals over 10 random seeds.

### L.2   GP SETTINGS

In GP settings (5.1), each expert demonstration involves 5 steps, is corrupted with Gaussian noise and collected by a scripted policy. As the task space is continuous, the criterion is simply optimized via uniform random shooting, with a budget of 100. Multi-task returns are averaged over 20 episodes per task.

### L.3   NEURAL NETWORK SETTINGS

#### L.3.1   ENVIRONMENTS

We evaluate AMF-NN across three environments, namely FrankaKitchen, Metaworld and WidowX. For the first two, demonstrations are 50 steps, while for the latter they involve 100 steps. In FrankaKitchen, demonstrations are provided by Kumar et al. (2024), and collected by trained SAC agents. In Metaworld, demostrations are instead collected by the scripted policies provided (Yu et al., 2020). Finally, in WidowX successful trajectories are collected by Octo-small (Octo Model Team et al., 2024) itself and filtered according to success labels, in an instance of self-supervised distillation. Furthermore, in the case of WidowX, the initial position of the object is not randomized, as we found this to result in very inconsistent performance for the data collection policy. In the first two

---

[4]sites.google.com/active-multitask-finetuning

environments, 25 attempts for each task are evaluated, while the evaluation for WidowX involves 50 attempts.

### L.3.2 BEHAVIOR CLONING

The policy is parameterized via a deterministic MLP with 2 layers and 256 units per layer, with layer normalization (Ba, 2016). Task conditioning are image embedding extracted by R3M (Nair et al., 2022a). Policies are pre-trained for 500 epochs with batch size of 128, learning rate of $1.e-4$ using the Adam optimizer (Kingma, 2014).

### L.3.3 IMPORTANCE SAMPLING WEIGHTS

In GP setting, importance weights are computed from the Gaussian policy distribution, and log-probabilities are clipped to the range $[-12, 0]$. In NN settings, for deterministic policies, we interpret the policy's output as the mean of a Gaussian with fixed standard deviation $\sigma = 1.0$, and only clip log-probabilities for numerical stability. In experiments involving pre-trained Octo policies, we evaluated two solutions. One option consisted of fitting a Gaussian distribution through maximum likelihood methods to samples from the diffusion policy, and was found to underperform. We thus treat the Octo policy as strictly deterministic: with continuous action spaces, this simplifies importance sampling weights to $w(c, \tau) = 1$ in case $\tau$ is a demonstration provided exactly for task $c$, and $0$ otherwise. We note that this solution cannot be used to evaluate the criterion on yet unobserved tasks, but remains feasible when tasks are finite and few.

### L.3.4 AMF

Each fine-tuning round involves 3000 gradient steps, each with a batch size of 128. We warm-start each algorithm by collecting the first $|\mathcal{C}|$ demonstrations uniformly, as mentioned in Section 4.2. In the case of loss-gradient embeddings, we found it to be beneficial to use a separate copy of the policy for task selection, which is not trained on these initial trajectories (which thus can be seen as a small "validation" set). As these demonstrations are not selected according to the criterion, to avoid unwanted updates of pre-training weights, they are only added to the training set once the algorithm selects a task belonging to the same partition of the task space, as defined by parameter isolation.

### L.4 RUNTIME

Each experimental run for AMF-NN takes at most 5 hours with GPU acceleration. In this case, data selection itself requires up to 8 minutes per round, and can be significantly sped up by reducing the sampling budget. AMF-GP experiments can be reproduced within 10 minutes on CPU.

## M USEFUL INEQUALITIES

**Lemma 8.** *If $a, b \in (0, M]$ for some $M > 0$ and $b \geq a$ then*

$$b - a \leq M \cdot \log\left(\frac{b}{a}\right).$$

*If additionally, $a \geq M'$ for some $M' > 0$ then*

$$b - a \geq M' \cdot \log\left(\frac{b}{a}\right).$$

*Proof.* Let $f(x) \stackrel{\text{def}}{=} \log x$. By the mean value theorem, there exists $c \in (a, b)$ such that

$$\frac{1}{c} = f'(c) = \frac{f(b) - f(a)}{b - a} = \frac{\log b - \log a}{b - a} = \frac{\log\left(\frac{b}{a}\right)}{b - a}.$$

Thus,

$$b - a = c \cdot \log\left(\frac{b}{a}\right) < M \cdot \log\left(\frac{b}{a}\right).$$

Under the additional condition that $a \geq M'$, we obtain

$$b - a = c \cdot \log\left(\frac{b}{a}\right) > M' \cdot \log\left(\frac{b}{a}\right).$$

$\square$

**Lemma 9.** *Let us consider two spaces $X \in \mathbb{R}^n$, $Y \in \mathbb{R}^m$, and a conditional distribution $p : X \to \Delta(Y)$ whose support $supp(p(\cdot|x))$ is bounded by a ball of radius $\epsilon$ for all $x \in X$, that is*

$$\max_{y_l, y_h \in \mathrm{supp} p(\cdot|x)} \|y_h - y_l\|_2 \leq \epsilon.$$

*For all $(x, x') \subseteq X$, $y \sim p(\cdot|x)$ and $y' \sim p(\cdot|x')$ it holds that*

$$\|y - y'\|_2 \leq \mathcal{W}(p(\cdot|x), p(\cdot|x')) + 2\epsilon,$$

*where $K$ denotes the Wasserstein 1-distance.*

*Proof.*

$$\|y - y'\| \leq \max_{\substack{y \in \mathrm{supp}(p(\cdot|x)) \\ y' \in \mathrm{supp}(p(\cdot|x'))}} \|y - y'\|_2 \tag{93}$$

$$\overset{(i)}{\leq} \min_{\substack{y \in \mathrm{supp}(p(\cdot|x)) \\ y' \in \mathrm{supp}(p(\cdot|x'))}} \|y - y'\|_2 + 2\epsilon \tag{94}$$

$$\overset{(ii)}{\leq} \mathcal{W}(p(\cdot|x), p(\cdot|x')) + 2\epsilon, \tag{95}$$

where (i) follows from the triangle inequality, and (2) is due to the fact that the integral of the distance between two points in $Y$ for any coupling is greater than the minimum distance. $\square$

