# OpenReview forum: "Active Fine-Tuning of Generalist Policies"
_ICLR.cc/2025/Conference — Submitted to ICLR 2025_

### Official Review · Reviewer_fnkz · 2024-10-23

**Soundness:** 3
**Presentation:** 3
**Contribution:** 2
**Rating:** 5
**Confidence:** 3

**Summary:**

The authors aim to apply active learning to a multi-task behavior cloning setup. In their problem setting, the agent is allowed to request demonstrations for the next task it wants to see a demonstration for. The expert policy provides this demonstration, and the goal is to define a strategy that maximizes expected return over a final evaluated task distribution (typically uniform over all tasks).

To do so, they use a Bayesian perspective, assuming that expert demonstrations have some independent sampled noise $\epsilon$, using a Gaussian process to model the policy. The method's goal is to maximize information gain per sample, given the history of prior trajectories + tasks. Under some assumption of Lipschitz continuity, they show the optimality gap can be bounded to decay in $O(infogain * n^{-1/2})$.

However, exactly optimizing the info gain for selecting a new task $c$ is not tractable as given, so we must instead use various approximations. Maximizing mutual info $\mathcal{I}(\pi(s_t,c); \tau(c'))$ over the next context $c'$ is the same as minimizing $\mathcal{H}(\pi(s_t,c) | c', \tau', history)$ over trajectories $\tau'$ sampled from the distribution of expert trajectories given task $c'$.

This is then made practical in two ways. First, to estimate sampling a new trajectory $\tau$, we apply importance weighting to the previous $n-1$ demos collected so far, using our current policy $\pi$ to generate importance weights. For entropy, since the policy is represented by a Gaussian process, we can leverage existing techniques from that literature for estimating entropy.

Last, to deal with forgetting, the authors apply "parameter isolation". The task-space is partitioned into $K$ sets, and $K$ copies of finetuning weights are initialized, each one only updated on tasks that lie within its partition.

In experiments, the authors find that active learning is most effective when the initial task distribution is skewed, with the gap between it and uniform sampling narrowing as the pretraining distribution becomes closer to uniform.

**Strengths:**

The authors provide a compelling justification for why active learning could be useful to robot learning, noting that several prior works use more bespoke methods for deciding on task distributions. The method is adjusted to be applicable to deep learning based policies, and shows promise, especially in scenarios with unbalanced tasks.

**Weaknesses:**

I find it somewhat concerning that parameter isolation (having separate weights per task) is so critical to final performance. My understanding was that existing models for Franka kitchen, Metaworld, etc. did not need to do this, and successfully trained models that could handle all contexts (tasks) at once.

The increase in performance from using AMF is relatively low in many settings, even the friendliest settings where task distributions are skewed. The gap often closes in skewed settings once we reach 10 rounds of data (see Figure 4).

With many ablations maxing out at a 20 demonstration budget, I am a little skeptical that the empirical importance weights are any good. It feels like 20 examples is too few to really get a good approximation of a high-dimensional trajectory space...

**Questions:**

I am curious about comparisons to other simpler sampling schemes. For example, something like "sample the context $c$ that appears least often in the history", or other similar schemes that use less machinery than estimating importance weights, info gain, etc.

---

> ### Author Response · Authors · 2024-11-19
>
> We would like to thank the reviewer for the thorough review, and their suggestions. We believe that we can address the outlined weaknesses, and provide additional empirical support to clarify our contribution.
>
> **Parameter isolation**
>
> Parameter isolation is critical because we tackle a relatively novel, more challenging setting. In the standard multi-task behavior cloning setting, all data that the policy is trained on remains available at all times, and the data distribution does not shift. In our setting, the policy is pre-trained on a sizeable pre-training dataset, but this dataset is *not available* during fine-tuning (see common response). This induces a large distribution shift between pre-training and early rounds of fine-tuning.  If this distribution shift is not addressed, performance suffers significantly, as shown in Figure 4 and Appendix G. Thus, the necessity of some form of continual learning arises from our problem setting itself, rather than from particular algorithms.
>
> An effective technique to prevent forgetting in a continual setting is rehearsal, which implicitly occurs in the standard multi-task setting, as the dataset is fixed and always accessible. Unfortunately, this is simply not possible when part of the training data becomes unavailable, as in our setting. For this reason, we benchmarked different continual learning techniques, and found parameter isolation to perform best (see Appendix G, including updates). We also note that, at the end of fine-tuning, it is possible to distill the sets of weights into a single, multi-task policy.
>
> **Importance weights**
>
> We appreciate this suggestion, and we are happy to add an analysis of importance weights as Appendix J. In short, for both AMF-GP and AMF-NN we confirm that they might be inaccurate in early parts of training. Nevertheless, we find that they become more reliable estimates in later rounds. We refer to Appendix J for a more complete discussion.
>
> **Other sampling schemes**
>
> Thank you for suggesting this baseline: we think it highlights a nice feature of the problem we tackle. We included a comparison in our revision. Please, see the common response for a detailed response, and further empirical evidence. In short, we find that our method matches or outperforms the performance of the suggested baseline, despite not having access to the pre-training data distribution.
>
> We hope that we were able to clarify the challenging fine-tuning setting we tackle, and why particular techniques might become desirable or not applicable to this case. In light of this, we would ask the reviewer to consider raising their score. We are of course happy to discuss and address any further questions or concerns!

---

> ### Author Response · Authors · 2024-11-23
>
> Dear reviewer,
> We greatly appreciate your time and effort in reviewing our work, and we believe that our response addresses the points you raised. As the discussion period is nearing its end, we wanted to kindly check if there is anything else we can clarify. We remain available for any further discussion.

---

> > ### Comment · Reviewer_fnkz · 2024-11-26
> >
> > Thanks for the comments. After reading them, I plan to keep my score the same.

---

### Official Review · Reviewer_Qd9g · 2024-10-30

**Soundness:** 3
**Presentation:** 4
**Contribution:** 2
**Rating:** 6
**Confidence:** 3

**Summary:**

This paper introduces AMF (Active Multi-task Fine-tuning), an algorithm for efficiently fine-tuning pre-trained "generalist" robot policies to perform multiple tasks. Given a limited demonstration budget, AMF actively selects which tasks to request demonstrations for, aiming to maximize overall multi-task performance.  It does this by selecting tasks that yield the largest information gain about the expert policy, focusing on areas where the current policy is most uncertain.

The authors provide theoretical performance guarantees for AMF under regularity assumptions, showing that it converges to the expert policy in sufficiently smooth MDPs. They also demonstrate AMF's effectiveness in practice, applying it to some robotic manipulation tasks with neural network policies. Experiments in simulated robotic environments like FrankaKitchen and Metaworld show that AMF significantly outperforms uniform task sampling, especially when the pre-training data is skewed towards a subset of tasks. The authors also demonstrated that AMF can be applied to off-the-shelf models like Octo, though the improvement over the naive baseline is marginal.

**Strengths:**

- The paper studies the timely problem of efficiently fine-tuning generalist robot policies, which are becoming increasingly important in robotics.

- The authors provide performance guarantees for AMF under certain regularity assumptions, proving its convergence to the expert policy in smooth MDPs. This adds to the credibility and understanding of the algorithm's behavior.

- AMF demonstrates improvements over uniform sampling, particularly when the pre-training data is biased towards a subset of tasks. This is an advantage as real-world pre-training datasets are sometimes unevenly distributed.

- The algorithm can be applied to some robotic environments with high-dimensional observation and action spaces, using neural network policies. This demonstrates its practical applicability in realistic scenarios.

**Weaknesses:**

### Major

- FrankaKitchen and MetaWorld are relatively simple robotic benchmarks due to their narrow initial state distributions and short task horizons of each task. Future evaluations would benefit from testing on more challenging robotic benchmarks such as RLBench [1], RoboSuite [2], ManiSkill [3], and BiGym [4], which offer greater complexity and variability.

- The effectiveness of AMF, especially with neural networks (AMF-NN), hinges on accurate uncertainty estimation. While the proposed loss-gradient embedding approach works well empirically, uncertainty quantification in neural networks remains a challenging open problem. The performance can degrade if the uncertainty estimates are unreliable.

- While AMF shows certain improvements in skewed pre-training scenarios, its advantage diminishes when the pre-training data is uniformly distributed across tasks. In such cases, simpler methods like uniform sampling might suffice.

- Evaluating the information gain criterion can be computationally expensive, especially with a large number of tasks or long trajectories.  The authors propose some approximations to mitigate this, but the computational cost can still be a concern in large-scale applications.

- While parameter isolation helps mitigate forgetting, it also prevents positive transfer learning across tasks. In addition, parameter isolation also adds more parameters to the final policy.

- The experiments with the Octo model are limited in scope, only fine-tuning the action head and focusing on a scenario with a relatively uniform pre-training distribution. More extensive evaluation with large-scale, generalist policies is beneficial to fully assess AMF's potential.


### Minor

- The points in Figure 2 are not well aligned.


### References

[1] James, Stephen, et al. "Rlbench: The robot learning benchmark & learning environment." IEEE Robotics and Automation Letters 5.2 (2020): 3019-3026.

[2] Zhu, Yuke, et al. "robosuite: A modular simulation framework and benchmark for robot learning." arXiv preprint arXiv:2009.12293 (2020).

[3] Mu, Tongzhou, et al. "Maniskill: Generalizable manipulation skill benchmark with large-scale demonstrations." arXiv preprint arXiv:2107.14483 (2021).

[4] Chernyadev, Nikita, et al. "BiGym: A Demo-Driven Mobile Bi-Manual Manipulation Benchmark." arXiv preprint arXiv:2407.07788 (2024).

**Questions:**

- In Figure 4, why does the red curve always drop in the first round? This phenomenon occurs in all tasks.

---

> ### Author Response · Authors · 2024-11-19
>
> We thank the reviewer for the detailed review and questions. We’ll comment on the weaknesses, and answer questions one by one.
>
> **Benchmarks**
>
> We appreciate the suggestion to explore more complex benchmarks, and agree that this represents a promising avenue for future research. We find that the underlying learning algorithm (in this case, behavior cloning (BC)) may not be able to solve very complex tasks, even in the single-task setting. To the best of our knowledge, the best BC baseline across the suggested environments only solves (>80% success rate) less than a third of available tasks. Thus, these environments largely test the behavior cloning component, which makes them less informative of the data selection technique. Our evaluation focuses on showing that, as long as behavior cloning *can* solve a set of tasks, AMF can substantially improve sample efficiency.
>
> **Performance after uniform pre-training**
>
> As the reviewer correctly points out, in the uniform settings, the performance gap between AMF and baselines shrinks. However, we argue that this does not occur because the performance of AMF worsens, but rather because this setting is particularly favorable to uniform task selection. The fact that AMF recovers its performance *without any knowledge of the pre-training distribution* confirms its ability to adapt across scenarios, and is a desirable trait. Moreover, we believe that a scenario in which the pre-training covers all tasks equally and sufficiently is rather unrealistic. In this case, fine-tuning might simply not be necessary. For any practical use of generalist policies, we can expect some substantial distribution shift, and this is the setting where AMF shows promising performance gains.
>
> **Runtime**
>
> We find the computational requirements for AMF to be modest. Even with a generous sampling budget, data selection with AMF-NN requires at most <8 minutes at each round, which represent a small fraction of the time required for the entire fine-tuning process. We have added details on runtime to Appendix N.4.
>
> **Parameter isolation**
>
> We agree with the reviewer: unfortunately, parameter isolation prevents positive transfer, but we found it to work best across several continual learning techniques, as shown in Appendix G. In our setting, the impact of positive transfer is overall lower than that of catastrophic forgetting. At the end of fine-tuning, the parameter count may be reduced by distilling the multiple copies of the weights into one.
>
> **Scope of OCTO experiments**
>
> To the best of our knowledge, experiments with OCTO include all currently available environments for Real-to-Sim. The evaluation is thus currently limited by lack of available simulated benchmarks. We agree that further experiments in this setting would be interesting, and look forward to the development of broader benchmarks for Real-to-Sim.
>
> **Figure 2**
>
> Thanks for pointing out the misalignment in Figure 2! As it was generated by plotting actual noisy demonstrations, the trajectories are not perfectly straight. We’re happy to tune down the noise for plotting purposes, and have replaced the Figure.
>
> **Figure 4**
>
> In Figure 4, early drops in the red curve are a symptom of catastrophic forgetting in neural networks. Across all tasks, networks are pre-trained to reach non-trivial performance in at least some tasks. This performance is shown by the first point for each curve. On the first round of fine-tuning, the fine-tuning dataset only contains the first selected batch of demonstrations, thus inducing a large distribution shift from the pre-training distribution. For this reason, the first round of fine-tuning “undoes” some of the pre-training, and performance for some tasks drops significantly. Hence, average performance also drops. This is a general issue in continual and lifelong reinforcement learning, which arises from the setting itself. In the context of this work, we adopt parameter separation to mitigate this issue, as described in line 348 and Appendix G.
>
> We thank the reviewer for the discussion, which highlighted some important features of our problem setting, and allowed us to clarify relevant points. Please, let us know if you have any further questions or concerns.

---

> ### Author Response · Authors · 2024-11-23
>
> Dear reviewer,
> We greatly appreciate your time and effort in reviewing our work, and we believe that our response addresses the points you raised. As the discussion period is nearing its end, we wanted to kindly check if there is anything else we can clarify. We remain available for any further discussion.

---

> > ### Comment · Reviewer_Qd9g · 2024-11-25
> >
> > Thank the authors for the detailed response.
> >
> > > Thus, these environments largely test the behavior cloning component, which makes them less informative of the data selection technique. Our evaluation focuses on showing that, as long as behavior cloning can solve a set of tasks, AMF can substantially improve sample efficiency.
> >
> >
> > Many modern imitation learning approaches (e.g., ACT, Diffusion Policy) can be viewed as fancy versions of behavioral cloning (BC). Given that naive BC with MLPs/CNNs often exhibit limited performance in complex robotics tasks, it is crucial to demonstrate that the proposed AMF method is compatible with modern imitation learning backbones capable of solving the above benchmarks.

---

> > > ### Author Response · Authors · 2024-11-28
> > >
> > > Thank you for further commenting on this issue. We understand that large scale evaluation would provide further support. On the topic of compatibility, AMF-NN only requires the policy class to be differentiable, such that loss gradient embedding may be computed. It is thus compatible with both ACT and Diffusion Policy; in fact, experiments with OCTO (Section 5.5) use a diffusion policy head.
> > > Please, let us know whether the other comments were sufficiently addressed in our previous response; we remain available to further discuss them.

---

### Official Review · Reviewer_gsEP · 2024-11-04

**Soundness:** 3
**Presentation:** 3
**Contribution:** 3
**Rating:** 6
**Confidence:** 2

**Summary:**

This paper studies an interactive multi-task fine-tuning setting, where the agent can adaptively select which tasks to demonstrate, towards fine-tuning pre-trained generalist robot policies. The authors propose Active Multi-task Fine-tuning (AMF), which maximizes multi-task policy performance by selecting demonstrations that result in the largest information gain on the expert policy. Statistical guarantees for AMF are proven by extending the results on active learning from (Hübotter et al., 2024) to the sequential decision making setting. Results are demonstrated in simulation, including for a transformer model pre-trained on a large-scale real-world robotic dataset.

**Strengths:**

- The proposed approach AMF builds in principled prior work in active learning, and performance guarantees are proven, assuming some smoothness and regularity conditions.

- The paper is generally well-written, and describes an approach from theory to practice.

**Weaknesses:**

- Two practical instantiations of AMF are introduced in the main paper, which parameterizes policies either using a Gaussian process or a neural network. Considering AMF-GP is only used for a simple 2D integrator task, and to help set up AMF-NN, I think it would be helpful to spend more space describing AMF-NN and moving some content for AMF-GP to the Appendix. Algorithm 1 could be replaced with a full description of AMF-NN, which also includes the design choices (kernel approximation, batch selection, parameter isolation) needed to obtain a working implementation.

- AMF-NN relies on parameter isolation to work (results in Figure 5), and does not yield benefits over uniform task selection baseline when finetuning a pre-trained robot transformer model in Section 5.5.

**Questions:**

L289: For AMF-NN, is Equation 2 optimized by also training a Gaussian Process? Could the conditional entropy be estimated by comparing samples from the initial pretrained policy?

L351-352: How is the task-space partitioning done?  Is a different set of weights used for each task? If so, it does not seem accurate to claim that AMF-NN can be scaled to large task spaces, particularly when fine-tuning larger models.

Model merging (https://arxiv.org/abs/2310.01362) could be an alternative solution to the strategies evaluated in Figure 14.

Figure 5, Figure 7: Why is parameter isolation ablation only done for uniform? How well does AMF perform without parameter isolation?

What tasks are selected by AMF over each round? Does this change over fine-tuning? Would be nice to see a plot of this alongside the success rates for each task.

---

> ### Author Response · Authors · 2024-11-19
>
> We thank the reviewer for the detailed review and questions. As suggested, we have edited our submission to further highlight AMF-NN, namely by editing Algorithm 1. For the time being, we have kept a concise introduction to GPs, as AMF-NN largely builds upon a GP approximation. We hope that this solution can improve the readability of our paper for readers that have a stronger background in both GPs and NNs. All other questions are answered one by one in the following.
>
> **Gaussian Processes in AMF-NN**
>
> In AMF-NN, no Gaussian Processes need to be explicitly trained. Instead, we leverage embeddings learned by the policy network, and a GP approximation to estimate the policy’s conditional entropy. If the pretrained policy also captured epistemic uncertainty, then entropy reduction could be estimated by comparing fine-tuned and pretrained policies. However, this process would require several gradient steps for each of the tasks to consider, while our current method remains more computationally efficient.
>
> **Task-space partitioning**
>
> In practice, we found that task-space partitioning for AMF-NN works best when having a set of weights for each task. As the partitioning gets coarser, performance gradually decreases. This limitation is entirely due to forgetting issues in neural networks, and we expect that improvements in continual learning would eventually remove it. Generally speaking, the AMF framework scales to continuous sets of tasks, as displayed by AMF-GP (which does not require any partitioning).
>
> **Model merging**
>
> We appreciate the suggestion regarding model merging. We find that its applications for continual learning [1,2] rely on the availability of a replay buffer of past data. In our case, as we further discuss in the common response, this is not possible: fine-tuning algorithms do not have access to pre-training data. However, we found that one of the methods at the core of model-merging (Git-ReBasin [3]) is also described in a “weight-matching” variant, which does not require data, and is therefore directly applicable in our setting. In the context of this rebuttal, we have implemented a continual learning algorithm based on this idea, as detailed in Appendix G. Unfortunately, we found it to be empirically ineffective in our fine-tuning setting.
>
> **AMF with no parameter isolation**
>
> Without parameter isolation, all methods we evaluated in Kitchen and Metaworld suffer from catastrophic forgetting, and perform equally. We only report curves for the naive baseline, as this drop in performance is independent of the data selection algorithm. Catastrophic forgetting is thus a challenge arising from our setting (in which the pre-training data is unavailable), and not from specific algorithms. As suggested by the reviewer, we have added complete plots as Figure 15.
>
> **Tasks selected over fine-tuning**
>
> We thank the reviewer for suggesting an analysis of task selection frequencies over training. We added this analysis in Appendix I, where we observe qualitatively interesting patterns. In short, we find that AMF is capable of rebalancing the training data distribution, but is also consistently selecting tasks based on learning progress or correlation.
>
> We thank the reviewer for suggesting several additional experiments. We believe that these additional results have been very helpful in improving our paper. We hope that our clarifications also address the remaining comments. We would be happy to discuss any further questions!
>
> **References**
>
> [1] Wang et al., “Robot Fleet Learning via Policy Merging”, ICLR 2024
>
> [2] Pena et al., “Re-basin via Implicit Sinkhorn Differentiation”, CVPR 2023
>
> [3] Ainsworth et al. “Git Re-Basin: Merging Models modulo Permutation Symmetries”, ICLR 2023

---

> > ### Comment · Reviewer_gsEP · 2024-11-23
> > **Remaining concern regarding parameter isolation**
> >
> > Thanks for answering my questions on AMF and for incorporating the suggestions to improve the presentation within the paper.
> >
> > To be precise, my main concern (one which other reviewers seem to share as well), is that your practical implementation of AMF-NN relies on parameter isolation. I would have raised my score further if the authors demonstrate (AMF-NN, no parameter isolation) outperforming (Uniform, no parameter isolation) in any of {FrankaKitchen, MetaWorld, WidowX} domains.
> >
> > Based on the results on Figure 15 it seems that without parameter isolation, AMF-NN is *no better or worse than uniform selection* in the FrankaKitchen and MetaWorld settings. Regarding the new changes within the appendix, I did not see results with (AMF-NN, no parameter isolation) for the Octo model on the WidowX tasks, see comment in my original review on Figure 7. This is concerning as a separate set of model parameters is used for each task. This effectively corresponds to overfitting on every single task (which is likely why AMF-NN w/ parameter isolation outperforms uniform selection), rather than adhering to the multi-task setting which this work targets.
> >
> > Observing catastrophic forgetting when fine-tuning on FrankaKitchen and MetaWorld tasks could be due to small network sizes (Appendix L.3.2). Larger models have been shown to be more robust to catastrophic forgetting [[Ramasesh et al., 2022](https://openreview.net/forum?id=GhVS8_yPeEa)], and simple weight averaging (based on the linear mode connectivity phenomenon) shows good performance in the multi-task fine-tuning setting of large multimodal models without requiring access to the pre-training dataset [[Ilharco et al., 2022](https://arxiv.org/abs/2208.05592)].
> >
> > As the paper is written from the perspective of proposing a better approach to fine-tune large multi-task models, I think this gap is a major shortcoming of the paper. Overall however, I still lean towards acceptance in the current state, as the paper is commendable for theoretically motivating an algorithm (for multi-task fine-tuning of models without access to the pre-training distribution) and providing some practical instantiation.

---

> ### Author Response · Authors · 2024-11-23
>
> Dear reviewer,
> We greatly appreciate your time and effort in reviewing our work, and we believe that our response addresses the points you raised. As the discussion period is nearing its end, we wanted to kindly check if there is anything else we can clarify. We remain available for any further discussion.

---

> ### Author Response · Authors · 2024-11-25
> **Comment on remaining concern**
>
> Thank you for supporting our submission, and providing more detailed and informative feedback. We understand your comments and would like to provide some more context.
>
> We would like to stress that parameter isolation is adopted as a solution to a problem, that is catastrophic forgetting. Unfortunately, without addressing catastrophic forgetting, fine-tuning a pre-trained model loses much of its nuances, as the pre-training may quickly be undone. As forgetting constitutes a bottleneck, it is unlikely that active data selection can be effective.
> We find this issue to arise particularly in the setting of Kitchen and Metaworld: in this case, parameter isolation happens to be the best performing solution. We agree that increasing model and dataset size can alleviate this issue, but we believe it is still important to highlight its occurrence with networks and dataset of modest size, and report the solution that was found to be most effective.
>
> One disadvantage of parameter isolation is that it prevents positive interference across tasks, and specializes a set of weights to each task, as the reviewer points out, and we acknowledge in lines 352-353. However, we would like to stress that the uniform baseline (e.g. Fig. 4, in yellow) also adopts parameter isolation. Thus, the better performance of AMF-NN cannot be explained by parameter isolation, and is rather induced by a different data selection strategy.
>
> As requested, we have further updated Appendix G to include results for OCTO without parameter isolation (new Fig. 15). In this setting, we find that AMF-NN without parameter isolation performs within confidence intervals of AMF-NN with parameter isolation. It is competitive with baselines, but does not significantly outperform them. As reported in lines 508-510, this is likely due to the fact that all fine-tuning tasks belong to the pre-training distribution, and a uniform sampling strategy is thus very desirable.
> However, these results suggest that parameter isolation is not strictly necessary for AMF-NN with large models, as scale partially alleviates catastrophic forgetting. This confirms two important points: (1) catastrophic forgetting is more prominent at smaller scales, and (2) it is a general issue in fine-tuning, and does not affect any strategy in particular.
>
> Finally, we thank the reviewer for stressing the importance of proposing principled fine-tuning techniques, and working towards practical instantiations, which we fully agree with.

---

> > ### Comment · Reviewer_gsEP · 2024-11-26
> > **(#2) Remaining concern regarding parameter isolation**
> >
> > Again, my concern is that AMF--the proposed approach for fine-tuning pre-trained policies in the *multi-task* setting--is using parameter isolation to effectively decompose the multi-task setting into several *single-task* fine-tuning problems using GT task partitioning. See my statement above: "This effectively corresponds to overfitting on every single task (which is likely why AMF-NN w/ parameter isolation outperforms uniform selection), rather than adhering to the multi-task setting which this work targets."
> >
> > Aside: Why does (Uniform, no parameter isolation) outperform (Uniform) in Fig. 15, and in Fig. 4 Kitchen Visual?
> >
> > From the updated Fig. 15, it looks like (AMF-NN, no parameter isolation) performs similarly to (Uniform, no parameter isolation). This means that the practical implementation of the proposed approach is *no better than the uniform baseline in the actual multi-task setting*, in the Octo/WidowX domain or on smaller scale models in FrankaKitchen and MetaWorld domains. Unfortunately, I cannot raise my score further given this limitation as of now. My original suggestions on model merging and other ideas were potential suggestions to overcome this issue.

---

> > > ### Author Response · Authors · 2024-11-28
> > > **(#2) Comment on remaining concern**
> > >
> > > Thank you for providing further feedback. We agree that parameter isolation is not an ideal solution to the catastrophic forgetting problem, although we empirically found it to be most effective. We hope that future works, perhaps on the line of model merging, will solve this issue, such that they can be integrated in an AMF framework. For the moment, we find that catastrophic forgetting remains an open problem, and focus our work on proposing a principled algorithm, and showing the extend to which it can scale.
> > >
> > > We are happy to answer the remaining questions:
> > > * in Fig.4 (Kitchen Visual), parameter isolation prevents a large drop in performance in the early stage of fine-tuning. Asymptotically, (Uniform, no parameter isolation) reaches similar performance to (Uniform): we hypothesize that this could be due to the limitations of parameter isolation, which prevents positive task transfer.
> > > * In Fig. 15, we also observe that (Uniform, no parameter isolation) matches or outperforms (Uniform). In this case, the scale of the policy could be largely addressing catastrophic forgetting (notice the absence of a performance drop in early fine-tuning). However, performance for all fine-tuning strategies is very similar in Fig. 15: in this setting, we do not claim that any of the solutions outperforms the others (see line 508).

---

### Official Review · Reviewer_fkov · 2024-11-04

**Soundness:** 2
**Presentation:** 3
**Contribution:** 2
**Rating:** 5
**Confidence:** 3

**Summary:**

This paper introduces an algorithm designed for the scenario where a pretrained multitask policy needs to be fine-tuned with a limited number of additional demonstration rounds, denoted as N. The algorithm focuses on maximizing sample efficiency by strategically selecting a task at each round for demonstration. It employs proxies for mutual information between the expert and the current dataset, proving convergence under certain assumptions regarding the noisy expert policy and the environment MDP. The paper presents multitask performance results in a simple 2D reaching environment and two simulated robot environments, demonstrating that the proposed method outperforms a uniform task selection strategy at each rounds.

**Strengths:**

- The paper includes proofs demonstrating that the fine-tuned policy converges to the expert policy under specific assumptions about the noisy expert and the dynamics of the environment, thereby offering theoretical performance guarantees.
- This work could be relevant in the context of large-scale pretraining, as it aids in making informed decisions about what data to collect or include in subsequent training iterations.

**Weaknesses:**

- The assumptions made about the expert policy and the noise in the expert policy significantly limit the method's applicability. In tasks involving physical robots, the expert policy is often humans which may not satisfy any of these assumptions.
- The paper only compares the proposed method against one baseline that uniformly selects tasks at each round, demonstrating superior performance, particularly when the policy is pretrained on a skewed composition of tasks. However, it remains unclear how the proposed method compares to other baselines, e.g. a baseline that naively chooses tasks at each round with the goal of balancing task distributions across the entire dataset, including the pretraining dataset. The specific advantages of the proposed method over this natural alternative approach are not addressed.
- In the simulated robot experiments, the individual tasks within each multitask suite are relatively disjoint. This raises questions about the proposed algorithm's effectiveness in scenarios where some tasks share structures, as simply balancing the trajectory count across tasks may not be the most effective strategy for leveraging the data collected thus far.

**Questions:**

- What is the trajectory count for each task, including those from the pre-training phase, at the end of each round when executing the proposed algorithm? Is the algorithm doing anything beyond simply balancing the trajectory counts across tasks?
-  Has it been proved that the simulated robot environments and expert policies satisfy the assumptions in Section 4.1? If not, what implications might this have for the performance of the algorithm?

---

> ### Author Response · Authors · 2024-11-19
>
> We would like to thank the reviewer for their thorough assessment. We believe that we can address all comments directly.
>
> **Assumptions**
>
> The assumptions in Section 4.1 are used to formally prove convergence *with high probability*, but they are not be necessary conditions for good performance. For instance, convergence still occurs under much weaker assumptions, albeit *in expectation* (see Appendix B). In high-dimensional cases, even these weaker assumptions may be violated. This is indeed the case in both Kitchen and Metaworld, particularly when learning from images. In these settings, we observe that AMF still performs well. While our theoretical results provide guarantees on AMF's performance in regular settings, we find it encouraging to empirically observe similar behavior in complex and high-dimensional tasks.
>
> **Rebalancing baseline**
>
> Thank you for this suggestion, we believe it highlights an important feature of our problem setting. Please, see the common response for a detailed response, and further empirical evidence. In short, we find that our method matches or outperforms the performance of this rebalancing baseline, despite not having access to the pre-training data distribution. Moreover, it displays data selection strategies that are more nuanced than simply rebalancing counts (e.g., it accounts for learning progress and correlations).
>
> **Shared structure across tasks**
>
> We agree that multi-task benchmarks generally involve relatively disjoint tasks. We selected evaluation environments specifically to include tasks with a significant amount of shared structure. In high-dimensional settings, while standard multi-task benchmarks often involve one scene per task, all tasks within our Kitchen or Metaworld evaluation share the same scene and initial state distribution (i.e., all objects are always present, irrespectively of the task). Thus, state distributions for different tasks are largely overlapping. Moreover, the expert demonstrations for several pairs of tasks will share some sequences (e.g., part of the demonstration for grabbing a cup and moving it to the left is also valid for moving it to the right). For this reason, we would argue that the amount of shared structure is significant, and larger than in many multi-task benchmarks.
> Additionally, experiments on the 2D integrator involve a continuous task space, in which two tasks might be arbitrarily close to each other, and their demonstration can be very correlated.
> The final evaluation on Simpler features arguably more disjoint tasks, although most of them involve picking and placing objects.
>
> We hope that our response clarifies our setting, and provides additional empirical evidence demonstrating that AMF goes beyond rebalancing the count of demonstrations. If this is the case, we would ask the reviewer to consider raising their score.
> We of course remain available for further discussion!

---

> ### Author Response · Authors · 2024-11-23
>
> Dear reviewer,
> We greatly appreciate your time and effort in reviewing our work, and we believe that our response addresses the points you raised. As the discussion period is nearing its end, we wanted to kindly check if there is anything else we can clarify. We remain available for any further discussion.

---

> > ### Comment · Reviewer_fkov · 2024-11-27
> >
> > Thank you for your rebuttal and for including the rebalancing experiment. Based on the new experiment results, I have decided to keep my score unchanged.

---

### Author Response · Authors · 2024-11-19
**General response**

We thank all reviewers for providing insightful comments. We are glad to hear that all four reviewers appreciated both the generality of the framework we propose, and the theoretical guarantees we derive.

In this general response, we would like to clarify one important point, and list additional results. We updated our submission accordingly; changes are highlighted in red. We refer to individual responses for a more detailed discussion, and we remain available for any further question.

**Clarification: access to pre-training dataset and rebalancing baseline**

We would like to thank reviewers *fkov* and *fnkz* for suggesting a “rebalancing” baseline, which would sample tasks that have been demonstrated the least in the history of training. We think that this suggestion highlights a very important feature of our fine-tuning setting: *we do not assume access to the pre-training dataset*. This feature is both practical, as this dataset might be unpractically large, and realistic, as releasing training datasets is even less common than releasing model weights (see, for instance, language models). This makes our setting more widely applicable, but also much more challenging.
As the pre-training dataset is not available, the suggested sampling scheme could only consider the history of demonstrations provided for fine-tuning, and would thus recover the performance of uniform sampling in expectation.

Nevertheless, we have implemented this criterion as a privileged baseline (Appendix H), granting access to pre-training counts. We observe that our method matches or exceeds its performance *without any knowledge of the pre-training distribution* (!). Furthermore, we also prepared a qualitative study of data selection with AMF (Appendix I), showing that our information-based criterion is indeed capable of rebalancing counts, but can additionally sample tasks that are “hard to learn” more often, and accounts for task correlation.

**Additional results**

We followed the reviewers’ suggestions, and gathered additional results that further support our contributions:
* We introduce and evaluate a privileged “rebalacing” baseline, as detailed above (Appendix H).
* We perform a qualitative study on single-task performance and data selection over the course of fine-tuning (Appendix I).
* We present an analysis of importance sampling weights in AMF-GP and AMF-NN, further describing how they are estimated (Appendix J).
* We extend Appendix G to include an additional continual learning technique, inspired by Reviewer *gsEP*.
* We further ablate parameter isolation in Appendix G, Figure 15.

---

> ### Author Response · Authors · 2024-11-28
> **Update to the general response**
>
> In light of the discussion phase, we would like to comment on a few important points:
>
> - We are glad to see the reviewers confirm their appreciation for the theoretical guarantees of our method (e.g. *"the paper is commendable for theoretically motivating an algorithm"*, gsEP), as well as its relevance to practical problems (*"this demonstrates its practical applicability in realistic scenarios"*, Qd9g).
> - We acknowledge that the particular instantiation of AMF-NN is limited by the issue of catastrophic forgetting, which we could only handle with hard parameter separation. This is not ideal in multi-task settings, as it prevents positive transfer for AMF-NN. Unfortunately, catastrophic forgetting remains a fundamental issue in neural networks, and is still far from being solved. As our main focus is proposing a principled algorithm, and studying how it can scale beyond regular setting, addressing catastrophic forgetting is not our main focus. We sincerely appreciate suggestions on this topic from Reviewer gsEP, and hope that further research in continual learning can provide more scalable approaches.
> - With the caveat of catastrophic forgetting, we provide evidence that AMF is applicable across policy classes (GP, MLP, OCTO), and can largely improve adaptation of policies to new tasks that were not primarily represented in the training distribution, with further evidence added as Appendix H and I.
>
> We again thank the reviewers for the discussion and remain available for further comments.

---

### Meta-Review · Area_Chair_MPgj · 2024-12-20

**Metareview:**

This paper introduces a method for actively selecting tasks to collect additional demonstrations during the fine-tuning of a generalist policy model, supported by theoretical performance guarantees under regularity assumptions.

Reviewers acknowledge the significance of the problem, the novelty of the approach, and the theoretical contributions. However, concerns are raised about the practicality of the method and significance of the results, particularly due to its reliance on parameter isolation to outperform a uniform baseline.

The authors made an effort to add experiments and show the proposed method can be implemented on different policy architectures. However, the performance gain of the proposed method over uniform baseline in the new experiment is marginal. I believe extending the results to realworld scenarios that could benefit from active task selection would greatly improve the impact of this work.

**Additional Comments On Reviewer Discussion:**

Authors clarified questions regarding assumptions and details of the algorithm, and added additional results during discussion.

---

### Decision · Program_Chairs · 2025-01-22

Reject